# Progress in Preparation of Sea Urchin-like Micro-/Nanoparticles

**DOI:** 10.3390/ma15082846

**Published:** 2022-04-13

**Authors:** Ruijing Ma, Liqin Xiang, Xiaopeng Zhao, Jianbo Yin

**Affiliations:** 1Smart Materials Laboratory, Department of Applied Physics, School of Physical Science and Technology, Northwestern Polytechnical University, Xi’an 710129, China; maruijing@mail.nwpu.edu.cn (R.M.); lqxiang@nwpu.edu.cn (L.X.); xpzhao@nwpu.edu.cn (X.Z.); 2Department of Physics and Electronic Engineering, Yuncheng University, Yuncheng 044000, China; 3Research and Development Institute of Northwestern Polytechnical University in Shenzhen, Shenzhen 518057, China

**Keywords:** urchin-like, micro-/nanoparticles, preparation method

## Abstract

Urchin-like microparticles/nanoparticles assembled from radial nanorods have a good appearance and high specific surface area, providing more exposed active sites and shortening the diffusion path of photoexcited carriers from the interior to the surface. The interfacial interaction and physical and chemical properties of the materials can be improved by the interfacial porous network induced by interlacing nano-branches. In addition, multiple reflections of the layered microstructure can absorb more incident light and improve the photocatalytic performance. Therefore, the synthesis and functionalization of three-dimensional urchin-like nanostructures with controllable size, shape, and hierarchy have attracted extensive attention. This review aims to provide an overview to summarize the structures, mechanism, and application of urchin-like microparticles/nanoparticles derived from diverse synthesis methods and decoration types. Firstly, the synthesis methods of solid urchin-like micro-/nanoparticles are listed, with emphasis on the hydrothermal/solvothermal method and the reaction mechanism of several typical examples. Subsequently, the preparation method of composite urchin-like micro-/nanoparticles is described from the perspective of coating and doping. Then, the research progress of urchin-like hollow microspheres is reviewed from the perspective of the step-by-step method and synchronous method, and the formation mechanism of forming urchin-like hollow microspheres is discussed. Finally, the application progress of sea urchin-like particles in the fields of photocatalysis, electrochemistry, electromagnetic wave absorption, electrorheological, and gas sensors is summarized.

## 1. Introduction

The physical and chemical properties of nanostructured materials are susceptible to size and morphology. For example, geometric morphology has a strong influence on the physical and chemical properties of titanium dioxide [1,2]. Controlling the morphology and size of titanium dioxide nanostructure has been proved to be the key to excellent photocatalytic photovoltaic and electrochemical properties [3]. In addition, the excellent performance of electrocatalysts is attributed to the synergistic effect of the structure and electronic structure of microspheres in the hydrogen evolution reaction and oxygen evolution reaction [4,5]. In particular, one-dimensional nanostructured arrays have become a hot spot in the field of electrocatalysis due to their large electrochemically active specific surface area and enhanced local electric field around nano-sized tips [6,7]. Therefore, increasing attention has been focused on the research of the control of structures, ensuring that the desired functional properties can be obtained.

In the past few years, one-dimensional nanostructures such as nanowires [8], nanorods [9], nanoribbons [10], nanotubes [11], nanowire arrays [12], and nanoflowers [13] have received increasing attention in many fields due to their unique properties. More recently, research is being extended to the fabrication of two-dimensional or three-dimensional ordered construction [14,15]. Three-dimensional (3D) architectures are self-assembled from one-dimensional nano-units, which not only have the properties of one-dimensional nanomaterials, such as quantum confinement effect, small-size effect, and surface effect [16,17], but also some new properties of three-dimensional assembly, such as quantum coupling effect and synergistic effect [18]. It has been found that various nanostructures with hierarchical systems can be prepared by adjusting reaction conditions, such as the core-shell structure [19,20], dendritic structure [21,22], urchin-like structure [23,24], and snowflake structure [25].

Urchin-like microparticles/nanoparticles assembled from radial nanorods have good appearance and a high specific surface area, which can provide more exposed active sites and shorten the diffusion path of photoexcited carriers from the interior to the surface. The interfacial interaction and physical and chemical properties of the materials can be improved by the interfacial porous network induced by interlacing nanobranches, and these pores can be modified later to add different functions to the materials [26]. In addition, multiple reflections of the layered microstructure can absorb more incident light and improve the photocatalytic performance [27]. Therefore, the synthesis and functionalization of three-dimensional urchin-like nanostructures with controllable size, shape, and hierarchy have attracted extensive attention. Currently, researchers are committed to exploring excellent synthesis methods for the preparation of urchin-like metal-oxides, hydroxide, sulphide selenide, telluride, phosphide, and hybrid materials, so as to extend the use of urchin-like micro-/nano structure in the fields of catalysis, capacity conversion, electromagnetic devices, electrode materials, sensors, biomedicine, absorbing and shielding materials, etc. [28,29,30,31,32,33,34].

The synthesis routes of urchin-like micro-/nanoparticles can be divided into two categories: direct growth and sequential branched modification. The resultant branched materials obtained by the direct growth method are completely embedded by epitaxial nanobranches of spiky or lamellar structure without the scaffold core. Such a route has its unique advantages. For instance, the branched nanostructures formed by spontaneous chemical reactions do not require complex processes or procedures to obtain complex hierarchies structures. Instead, this enables size control from nanostructures to the microscale [35]. However, the sequential surface-supported modification method is to assemble one-dimensional nanorods, nanowires or two-dimensional nanosheets onto the backbone to form a three-dimensional interconnected and interweaved network. Although the process is complex and time-consuming, such a route reduces the accurate control of the size and density of additional branches on the scaffolds, which are more flexible and direct for making three-dimensional complex branch structures.

In recent years, urchin-like micro-/nanoparticles have developed rapidly in various fields, but so far, there are few reviews on this aspect. In this review, we summarize and describe the synthesis, application, and reaction mechanism of urchin-like micro-/nanoparticles with controllable structure and morphology. First, the preparation of single urchin-like micro-/nanoparticles by hydrothermal, thermo-solvent, electrochemical, and microwave-assisted methods is described. Then, the preparation method of composite urchin-like micro-/nanoparticles is presented. After that, the research progress of the preparation of urchin-like hollow microspheres is reviewed. Finally, the applications of urchin-like particles in photocatalysis, electrochemistry, electromagnetic wave absorption, electrorheological fluids, and gas sensors are summarized. It is expected that this review will provide an overview of previous contributions and a certain understanding of the morphology, preparation, and mechanism of urchin-like micro-/nanoparticles, so as to fully exploit the potential of urchin-like micro-/nanoparticles in various fields.

## 2. Preparation of Urchin-like Micro-/Nanoparticles

### 2.1. Preparation of Solid Urchin-like Micro-/Nanoparticles

Understanding the formation mechanism of nanoparticles is the key to controlling their morphology. We have listed the preparation of urchin-like micro-/nanoparticles by hydrothermal, thermosolvent, electrochemical, and microwave-assisted methods, as well as their reaction mechanisms. It is expected that the potential of sea urchin-like micro-/nanoparticles in various materials can be fully developed.

#### 2.1.1. Solvothermal/Hydrothermal Growth

(1)Two-step method

In hydrothermal and solvothermal reactions, precursors are often used to synthesize urchin-like micro-/nanoparticles of different sizes. Then, the desired urchin-like compounds are obtained by calcination, vulcanization or phosphating. Metal salts are most commonly used because they dissolve easily. The size and monodispersity of urchin-like micro-/nanoparticles can be adjusted by adjusting reactant parameters (such as temperature, time, pressure, solvent properties). It should be noted that the rapid growth of particles in a highly polar solvent often leads to the formation of aggregates of large particles. Therefore, in order to control the uniformity and dispersion of urchin-like particles, the precursors are synthesized with the assistance of template surfactants. 

Template-assisted precursor conversion method

The growth mechanism of the surfactant-based templating method is that structural assembly occurs when the surfactant concentration exceeds the critical micelle concentration. In order to reduce the free energy of the system, the structures gather together by agglomeration for mutually permeable growth. Then, urchin-like nanoparticles are obtained by the Oswald mature welding structure under the regulation of the surfactant, capping agent, and guiding agent. Generally, different types of surfactants should be selected according to the needs of different chemical substances, but the proportion of surfactants in the reaction solution largely determines the structural properties of the obtained substances. For example, Fei et al. [36] prepared a 3D sea urchin-like precursor in aqueous solution with the help of cetyltrimethylammonium bromide(CTAB) surfactant by a simple and environmentally friendly high-temperature hydrothermal method and then converted it to urchin-like Co_3_O_4_ by calcination without the change of the urchin-like structure. With the formation of the urchin-like structure, it retained all the advantages of the nanowire, such as a high surface-to-volume ratio and short transmission length. In addition, the large porosity of the urchin-like structure also resisted the large volume expansion during the cyclic voltammogram (CV) and improved the capacity retention ability.

In aqueous media, Huang et al. [37] prepared bimetallic phosphating CoNiP with a novel urchin-like hierarchical structure composed of nanowires by using the precursor conversion method and used Co(NO_3_)_2_, Ni(NO_3_)_2_·6H_2_O and urea as raw materials with the aid of surfactant polyethylene glycol(PEG). The diameter of the urchin-like CoNiP was about 2~5 µm. The length of the nanowires was about 2 µm, and the diameter was about 50~100 nm. The author explored the reaction mechanism of the process in Figure 1. Polyethylene glycol is a long-chain non-ionic surfactant with hydrophilic ether oxygen radicals (-O- radicals). When Co(NO_3_)_2_ and Ni(NO_3_)_2_ solutions were added to the polyethylene glycol solution, CO^2+^ and Ni^2+^ interacted with oxygen free radicals and were adsorbed on the long chain of polyethylene glycol. The OH^−^ released by urea hydrolysis at high temperature caused CoNi(OH)_4_ to crystallize and nucleate. Due to the competitive deposition of bimetallic CoNi, the growth of the CoNi(OH)_4_ crystal around the PEG long chain was limited, which gradually grew into a nanowire structure. Interestingly, it was found that the hierarchically urchin-like structure not only provided a larger surface area and more active redox reaction sites, but also had an open structure that can promote rapid mass transfer. In addition, under low-voltage conditions, a sharp-tip enhancement effect was induced at the tip of the nanowires, resulting in a local high-voltage field around the nanowire tip, which lead to nanoscale free convection, improving mass transport. 

In general, different surfactants are selected according to different solvents. When alcohol was used as the reaction solvent, Sun et al. [38] selected sodium dodecyl sulfonate as a surfactant and InCl_3_·4H_2_O and urea as raw materials to prepare urchin-like precursors by the solvothermal method. Then, the precursors were loaded into an alumina tank and calcinated slowly to 500 °C at a rate of 2 °C/min for 2 h. Finally, urchin-like In_2_O_3_ with a diameter of 1 µm was obtained.

Considering the practical application and environmental protection, it is imperative to use green materials in the material preparation process. Therefore, the research team set out to develop strategies based on cheaper and less dangerous compound replacements. For example, Tong et al. [39] used the green glucose-guided hydrolythermal method to obtain an α-FeOOH precursor with a size of about 1 μm, and then urchin-like Fe_2_O_3_ nanoparticles were prepared via heat treatment, which had controllable morphology and crystallinity. A glucose molecule with 5 hydroxyl groups is easy to form a linear structure by the interaction of hydrogen bonds to promote α-FeOOH nanoparticles assemble into fiber bundles. At the same time, it also restricts the growth of the flake nanoparticles and produces a mass of pores with an average pore size of 6.94 nm. Finally, due to the synergistic effect of polar molecular interaction and residual magnetic moment interaction, the fiber bundles self-assembled into urchin-like nanostructures. At the same time, it is proposed that different heat treatment temperatures not only adjust the morphology, grain size, and crystallinity, but also adjust the pore structure, which is beneficial to the adsorption and transportation of chemical reagents.

Tong et al. [40] later used the same method described above and found that when the calcination temperature was raised from 300 °C to 400 °C in H_2_/Ar (1:4 volume ratio) mixing gas atmosphere, the urchin-like α-FeOOH precursors changed from Fe_2_O_3_ to Fe_3_O_4_. Interestingly, their building-blocks changed from fiber strands to rod-like nanoparticles. The authors explained this was because at higher temperatures (350–400 °C), nanoparticles grew dramatically by melting adjacent nanoparticles, leading to the formation of nanorods, whereas only rod-like α-Fe may be obtained at a relatively high temperature (500 °C). In addition, the author also analyzed the relationship between the dielectric constant and reduction temperature. Results showed that α-Fe_2_O_3_ obtained at T_r_ = 300 °C had enhanced dielectric loss, which was mainly attributed to the enhanced orientation and interface polarization in the urchin-like structures. Urchin-like Fe_3_O_4_ obtained at T_r_ = 350 °C exhibited significantly increasing ε′ and ε″. This is because the space charge polarization of the inverse spinel-type crystal with an urchin-like nanostructure led to a significant increase in dielectric relaxation loss, which improved the conductivity of Fe_3_O_4_, and thus improved the storage and dissipation capacity of electric energy.

b.Template-free precursor conversion method

Through the template method, the morphology, porosity, and texture of the sample can be well controlled, but at the same time, impurities will be introduced and the cost will be increased. Therefore, researchers are committed to exploring the template-free precursor conversion method. This method often selects urea as raw material because the hydroxyl and carbonate produced by the hydrolysis of urea in hydrothermal reaction not only participate in the formation of precursors and promote anisotropic growth, but also act as coordination agents, which is conducive to creating a more complex hierarchy. The most important thing is to not introduce impurities. Yuan et al. [41] demonstrated a simple and facile approach to fabricate urchin-like NiO architectures, which required neither template nor surfactant. Firstly, the urchin-like nickel precursor was prepared by the hydrothermal method using NiCl_2_·6H_2_O and urea as raw materials. The products were microspheres with a particle size of 1 µm, which were further composed of numerous one-dimensional dense nanowires that grew radially from the center. The nanowires were not uniform along the length direction, and their diameters were between 10 and 30 nm. Finally, the obtained nickel precursor was calcined at 400 °C in air for 2 h to obtain the urchin-like NiO. In order to study the formation process of the urchin-like structure, the authors collected some intermediates during the formation process of the precursor. According to the experimental results, the formation mechanism of the urchin-like precursor in the absence of a template and surfactant was proposed. The authors stated the formation of urchin-like architecture included two steps: the formation of a solid nucleus and growth of nanowires on the surface of nanoparticles. Initially, the Ni^2+^ concentration was higher. Urea decomposed and reacted with nickel ions to form nuclei, but these nuclei quickly coalesced to large spheres before they had time to grow. In the second step, NH_3_ produced during thermal decomposition of urea played an important role, which could help regulate the formation direction. Moreover, the product retained the new sea urchin-like structure after calcination.

Unlike Huang [37], Wang et al. [42] prepared a Co(CO_3_)_0.5_(OH)·0.11H_2_O precursor by the hydrothermal method. They did not add any surfactant, but dispersed Co(NO_3_)_2_ and urea in deionized water, and then urchin-like CoP was prepared by an in-situ phosphating process. Its size was 5 µm. The atomic ratio of Co/P was 1:1. The influence of the urea concentration on particle morphology was studied. It was found that hydroxyl and carbonate ions produced by urea hydrolysis not only participated in the formation of the precursor and promoted anisotropic growth but also facilitated the formation of more complex hierarchical structures at higher concentrations.

Jiang et al. [43] also synthesized a NiCo_2_S_4_ urchin-like nanostructure of ternary nickel-cobalt sulfide using urea as a raw material through the precursor conversion method without adding surfactants. Ni- and Co-based precursors with urchin-like structures were first synthesized by a simple hydrothermal route. In this process, Ni^2+^ and Co^2+^ were provided by NiCl_2_ and CoCl_2_, water was used as a solvent, and urea hydrolysis provided carbonate and hydroxyl ions. The XRD pattern of the precursor showed that since only a few Ni ions replaced Co ions, the lattice parameters were slightly changed, but the lattice structure was not changed. Then, in a hydrothermal environment, the nanostructure was in situ chemically transformed into NiCo_2_S_4_ with a Ni, Co, and S ratio of 1:2.06:3.91 by reaction with sodium sulphide (Na_2_S), and the morphology and structure of the precursor were completely maintained. NiCo_2_S_4_ was composed of different nanoribbons, which grew radially from the center.

(2)One-step method

By changing the hydrothermal/solvothermal synthesis conditions and post-treatment, different phase composition, crystallinity, and morphology can be obtained from the same type of precursor. This provides a feasible method for the accurate synthesis of urchin-like micro-/nanoparticles. This strategy also opens the door to other forms that cannot be achieved. However, the synthesis method discussed above includes multiple continuous steps, so it becomes cumbersome, complex, and time-consuming, with a high cost. In order to simplify the synthesis process, Li et al. [44] developed a simple hydrothermal method for large-scale synthesis of a uniform urchin-like γ-MnS system by a one-step method without using any surfactants and templates. As shown in Figure 2a–c, the sample was composed of a uniform urchin-like γ-MnS architecture with a diameter of about 4–5 µm. The urchin-like structure was composed of many nanorods with a diameter of 80–100 nm and length of 2–2.5 µm. The end face of the nanorods was hexagonal. A possible growth mechanism has been proposed as shown in Figure 2d. L-cysteine molecules have several functional groups that have a strong tendency to coordinate with inorganic cations and metals, such as -NH_2_, -COOH, and -SH. Firstly, in the solution, Mn^2+^ fully mixed with L-cysteine to form a Mn-cysteine complex, resulting in the formation of nanosheets. Under high temperature and high pressure, the C-S bond of L-cysteine broke, and the released S^2−^ anion combined with Mn^2+^ cations to form γ-MnS crystal particles. Due to the high surface energy, γ-MnS crystal particles were unstable at first, and they tended to aggregate into spheres driven by the minimal interfacial energy. Finally, driven by the inherent directional growth habit of γ-Mns, one-dimensional nanorods gradually formed on the surface of spherical particles and gradually became longer. In addition, the authors found that the solvent played an important role in the final morphology of the crystal. The mixed solution of deionized water and ethylene glycol (EG) could increase the reaction rate, which was conducive to the formation of a three-dimensional urchin-like structure. A separate solution or EG solution was not conducive to the formation of the three-dimensional structure. It is worth mentioning that the urchin-like γ-MnS system showed strong quantum size effects in UV-visible absorption tests. The photocatalytic investigation showed that the prepared urchin-like γ-MnS system had a good photocatalytic activity, which might have broad application prospects in industrial wastewater treatment.

Tang et al. [45] also proposed a similar growth mechanism and reported the synthesis of an urchin-like Bi_2_S_3_ nano/micron structure via the hydrothermal method. The material characterization showed that it was a uniform spherical symmetric urchin-like material with a diameter of 6–8 µm. The porous nanorods about 3–4 µm in length and 50 nm in diameter radially grew from the center of the long nanostructure of sea urchins in all directions. HRTEM images of single nanowires confirmed that the porous nanorods were polycrystalline and composed of nanosheets. The authors suggested that the self-assembly and intrinsic splitting characteristics of Bi_2_S_3_ structure were the reasons for the formation of the urchin-like structure. In the reaction process, the reaction time, reaction temperature, and thiourea were the key factors.

Although the above hydrothermal method is simple and the reaction time is greatly reduced compared with the two-step method, the thiourea required in the reaction is a carcinogen, and toxic gases such as nitrogen and sulfur oxides are released during thermal decomposition. Sang et al. [46] used a hydrothermal method to prepare a large number of urchin-like Bi_2_S_3_ structures without thiourea and other carcinogens in the raw materials. The process was simple and environmentally friendly. At the same time, the author also explored the effect of the nanostructure on the photocatalytic reduction of Cr under visible light. The results showed that the kinetic constant of urchin-like Bi_2_S_3_ nanostructures for Cr (VI) degradation was about 100 times that of Bi_2_S_3_ nanoflowers and commercial P25, and it was a highly efficient photocatalyst.

Han et al. [26] synthesized the urchin-like NiFeP photocatalyst by a one-step solvothermal method without any organic additives. It exhibited a high specific surface area and considerable mesoporous distribution, which provided more active catalytic sites for photocatalytic water oxidation and multi-electron transmission channels. Moreover, by comparing the XRD, FT-IR, and SEM data of the urchin-like NiFeP photocatalyst before and after the water oxidation reaction, the results showed that the structure and morphology of the photocatalyst did not change significantly, and the high oxygen yield of the catalyst was maintained even after the three recycling experiments. These results clearly indicated that the urchin-like NiFeP photocatalyst was highly stable for reuse. 

The morphologic changes of urchin-like structures can be realized by changing easily controlled technological parameters in the hydrothermal process, such as the reaction time, reactant concentration, pH value, and so on. Researchers have also explored this. For example, Luo et al. [47] synthesized urchin-like NiCo_2_O_4_ by a one-step hydrothermal method. The effects of the optimum synthesis conditions on the morphology and electrochemical properties of the samples were investigated by orthogonal experiments and single factor hydrothermal time. The synthesis conditions for preparing NiCo_2_O_4_ with different morphologies were systematically studied. Figure 3a shows the SEM of NiCo_2_O_4_ at different hydrothermal times. Its morphology and characteristics were urchin-like, flower-like, sheet-like, and block-like. As the reaction progressed, the needle-like material slowly extended outward, forming a flower-like sphere composed of uniformly dispersed flakes. These flower pieces continued to grow, which was not enough to cross-bond into a flower shape to form the nanosheet. As the nanosheets grew and thickened, they eventually formed a stable block. From the time of formation, urchin-like material was formed first, followed by flower-like, and flakes to a stable block. Among them, the transition time from the flower-like to flakes was the shortest, indicating that the flower-like was the most unstable state. The authors explained that the surface energy of the edge of the nanosheet that made up the flower-like spherical was relatively large, and the growth of the crystal proceeded in a stable direction.

Using strontium nitrate and urea as raw materials, Wang et al. [48] synthesized urchin-like strontium carbonate SrCO_3_ with a hierarchical nanostructure by a hydrothermal method, whose surface was composed of nanorods with a diameter of about 50 nm and a length of 12 µm that radiated outward from the center of the sphere. It was also found that the size and morphology of the product were affected by the change in the hydrothermal reaction time and the microconcentration of reactants as shown in Figure 3b. Urea was decomposed at high pressure in a closed container to produce CO_2_, which was converted into carbonic acid in deionized water. Then, Sr^2+^ reacted with CO_3_^2+^ to generate SrCO_3_ microspheres, which were self-assembled and expanded into spherical aggregates by small primary SrCO_3_ nanoparticles. In the following 6 h, the aggregates grew into dispersed single spheres gradually, and small particles in the solution were constantly adsorbed onto these spheres and grew into prototype nanorods. Finally, the nanorods became more uniform and elongated, diverging outward from the center of the sphere to form better urchin-like SrCO_3_ particles.

Due to the large specific surface area, the urchin-like microspheres provide more exposed active sites and shorten the diffusion path of photoexcited carriers from the interior to the surface. At the same time, interfacial interactions are promoted by interlacing nano-branched porous networks, which are commonly used as electrodes in electrochemical devices. Through hydrothermal reaction, the 3D array structure of urchin- like micro-/nanoparticles can be grown on various substrates, such as carbon cloth, metal foam, mesh, foil, and glass (FTO). The substrate not only improves the mechanical stability but also improves the electron/ion transport rate as a continuous conductive network. Therefore, the electrode mode of urchin-like micro-/nanoparticles deposited on the substrate has been widely used in flexible supercapacitors, electrochemical splitting water, batteries, and other energy storage converters. For example, Wang et al. [49] prepared uniform Zn nanostructures on the surface of copper foil by electrodeposition in an electrolyte, which consisted of 0.40 M HBO_3_, 2.55 M KCl, 0.44 M ZnCl_2_, and 2 g/L gelatin, and then synthesized urchin-like ZnO nanostructures by a simple hydrothermal method. FESEM images in Figure 3c clearly showed that a large number of conical ZnO nanorods radiated outwards from the center of the nanostructure, forming a spherical urchin-like structure with a diameter of 4–5 µm. The conical rod had an average diameter of 0.1–1.0 µm and a length of 3–5 µm. It was found that the hydrothermal time was a key factor affecting the surface morphology of ZnO nanostructures (Figure 3c). The highly ordered urchin-like ZnO nanostructures showed excellent photocatalytic properties.

Liu et al. [50] achieved morphology control of urchin-like V_2_O_5_ via an additive-free oil bath approach with subsequent thermal treatment. The urchin-like V_2_O_5_ microsphere had a diameter of 2~3 µm, and each microsphere was assembled by abundant nanorods. With different concentrations of raw materials, the nanorods and state of aggregation gradually changed (as shown in Figure 3d).

Shakir et al. [51] fabricated an urchin-like hierarchical Co_3_O_4_ nanoarchitecture directly on Ni-foam (NF) by a scalable hydrothermal and post-sintering treatment. The electrochemical structure was studied on the surface. Nickel foam-supported urchin-like stacked nanostructures were a promising high-performance electrode material for confluence supercapacitors whose excellent properties were attributed to the urchin-like open structure.

Above all, urchin-like monobasic metal oxides are directly grown on the substrate as electrodes, showing excellent properties. Due to the synergistic effect of different metal cations and various metal oxidation states, binary metal compounds will show ultra-high specific capacitance and excellent rate performance. Wang et al. [52] used a two-step hydrothermal method to directly grow urchin-like MnCo-selenide nanowires on the surface of nickel foam. The unique hierarchical microstructure, synergetic effect, and excellent conductivity enabled the electrode to exhibit outstanding super-capacitor performance. Qi et al. [53] used a simple and green two-step method to synthesize ternary metal oxide ZnMn_2_O_4_. It had the advantages of greater redox reaction and higher specific capacity.

Hua et al. [54] dropped FeCl_3_ solution on Si substrate and then kept it at a constant temperature of 400 °C for 1 h to deposit the urchin-like Fe_2_O_3_ superstructure on the silicon substrate. It was found that the morphology of the particles could be controlled by adjusting the temperature.

The most obvious advantage of in-situ growth of urchin-like micro-/nanoparticles on the external template is that the size and morphology of the obtained electrode can be directly controlled. However, before the reaction, the template substrate needs to be pretreated, such as polishing or pickling to remove the oxide layer and to increase the affinity with the reaction solution, so as to obtain a uniform coating.

#### 2.1.2. Other Methods

The solidification rate is an important factor for the morphology of the particles. Rapid solidification can form dendrites with sharp tips, and higher reduction temperature can accelerate the solidification rate of the particles. It is difficult to achieve high growth rates in aqueous solution at atmospheric pressure. Therefore, the reductions mentioned above are usually carried out at high temperature and in an autoclave [55]. Therefore, people continue to explore a simple and mild method to prepare sea urchin-like micro-/nanoparticles. Katsuyasu et al. [56] demonstrated a simple and facile method to fabricate urchin-like Ni architectures. The reduction of nickel ions proceeded at atmosphere pressure and was accelerated upon the addition of sodium carbonate (Na_2_CO_3_) to achieve the condition for growing needle-like structures. The authors believed that the presence of sodium carbonate did not regulate the pH value of the solution as previously thought, but acted as a catalyst or foaming agent.

Zhang et al. [57] synthesized urchin-like CdSe nanostructures with a diameter of 2.5–3.5 µm by a facile and mild W/O microemulsion approach assisted by surfactants, and the average diameter of nanorods was 100 nm and the length was 1 µm. The X-ray powder diffraction pattern of the product showed that it was pure CdSe with a zinc blende structure rather than the thermodynamically favored wurtzite structure.

In addition, Ballesteros et al. [58] reported that urchin-like zinc oxide nanostructures could be grown on galvanized sheets under UV irradiation and heat treatment at 500 °C for 2 h. It was found that the catalyst had certain application potential in dyes.

Zhang et al. [59] proposed a simple and controllable low-temperature chemical reduction method to assemble high-purity nickel hierarchical structures using spear-shaped nickel nanocrystals. By adjusting the reaction conditions and reaction solution components appropriately, sea urchin-like nickel particles and barbed nickel chains were selectively prepared at room temperature. The authors investigated the effects of the reaction time, reaction temperature, alkali concentration, and other parameters on the product morphology. The magnetic properties of the products were studied, and the results showed that the magnetic properties of the products were related to the corresponding microstructures. Based on a series of comparative experiments, a possible growth mechanism was proposed.

In the methods discussed above, the preparation process requires high temperature and a long reaction time. Although the electrochemical method can reduce the reaction time, it still takes several tens of minutes. Xiang et al. [60] put forward a simple microwave-assisted solvothermal method using TiCl_4_ aqueous solution as a reactant and toluene as a solvent to obtain urchin-like rutile TiO_2_ particles in just a few minutes without further heat treatment. The effects of temperature, irradiation time, and the reactant and solvent ratio on particle morphology and crystal structure were systematically studied. The author considered that microwave-assisted reaction can shorten the reaction time. On the one hand, microwave radiation heated the crystal titanium dioxide rapidly and uniformly. On the other hand, local overheating caused by specific microwave absorption of the polar component of the reaction (titanium dioxide) made the reactants more active under microwave irradiation than heating reaction. 

In addition to the above inorganic urchin-like micro-/nanoparticles, the researchers also developed different methods to prepare polyaniline (PANI) organic sea urchin-like micro-/nanoparticles. Choi et al. [61] self-assembled one-dimensional PANI nanowires prepared by oxidation polymerization into porous urchin-like spherical PANI microspheres with an average diameter of 2.5 µm. The density was 1.16 g/cm^3^, and the specific surface area was as high as 24.5 m^2^/g, which had typical electrical responsive characteristic behavior. Shen et al. [62] chose lignin sulfonate (LGS) as the template to synthesize sea urchin like PANI conductive microspheres. Cheng et al. [63] reported a simple template-free method for the synthesis of polyaniline (PANI) microspheres with an urchin-like structure. The surface of the microspheres was composed of nanofibers with a diameter of about 100 nm and a length of about 2 µm. The cyclic voltammetry and galvanostatic charge/discharge tests of microspheres in 1 M H_2_SO_4_ solution showed typical electrochemical characteristics. It was also proposed that the key reaction parameters such as the aniline concentration APS and aniline molar ratio affected the morphology and electrochemical capacitance of PANI microspheres.

Due to the anisotropic distribution of electromagnetic fields on the surface of non-spherical noble metal particles, they show unique surface-enhanced Raman spectroscopy (SERS) and catalytic behavior compared with spherical particles. In particular, both the theoretical calculation and experimental results show that there is a large electromagnetic field enhancement in the tips of branched particles, resulting in stronger SERS activity than in non-branched particles. Therefore, in recent years, reports on the synthesis of gold particles with different sizes and more branches have been emerging.

Kuo et al. [64] used sodium dodecyl sulfate (SDS) as a surfactant to prepare sea urchin-like gold nanoparticles with a particle size of 40 nm by a multi-step seed growth method. The particles prepared by this method can only be weakly stabilized by ligands electrostatically attracted to the metal core. Thus, all of these particles would coalesce irreversibly upon drying. In order to improve the stability of sea urchin-like gold nanoparticles, Bakr et al. [65] proposed a method for high-yield synthesis of sea urchin-like gold nanoparticles using MH, MUA, MPA, and these four thiol binary mixtures as capping agents. The nanoparticles were 40 nm in diameter, and due to the thiolated ligand shells that improve the solubility and stability of the particles, these particles were stable and water-soluble, and they can be stored in a dry state for several days and re-dispersed many times. After drying, the particles can be re-dispersed back in water and remain stable in solution with the exception of pure MH-coated nanoparticles. However, they re-dispersed in an aggregate form. Unfortunately, the authors did not elaborate on the mechanism of the formation of urchin-like gold nanoparticles. Zhang et al. [66] used the mixture of sodium citrate and hydroquinone as a reducing agent and demonstrated seed-mediated growth of urchin-like gold NPs by modulating the reactivity of gold ions, making it possible to adjust the diameters of the prepared NPs from 55 nm to 200 nm. The structure and size of branched particles were fine tuned. Significant progress has been made in controlling the number and length of branches. The reaction mechanism is also discussed in detail. The formation of urchin-like gold NPs includes two processes: gradual reduction of Au^III^ to Au^I^ and gradual reduction of Au^I^ to Au^0^. In the first step, a mild reducing agent citric acid, was used to reduce Au^III^ to Au^I^. In the second step, hydroquinone with high selectivity in reducing Au^I^ was selected to reduce it to Au^0^. This was because the standard reduction potential was 1.002 V in the presence of seeds, whereas it was −1.5 V in reducing isolated Au^I^ to Au^0^. Therefore, hydroquinone leads to preferential reduction of Au^I^ to Au^0^ on the seed surface to enhance the reactivity of gold ions. High concentrations of hydroquinone lead to excessive Au^0^ in the reaction system, which promoted rapid deposition of Au^0^ in the highly active (111) plane through a dynamically favorable process, resulting in branching growth. It is worth mentioning that the morphology of the prepared NPs was determined by the amount of hydroquinone added the first time and was independent of the total amount of hydroquinone because only the instantaneous high concentration of Au^0^ contributed to the preferential deposition of Au^0^ at the high energy surface. In addition, the urchin-like NPs prepared by this method had good stability in aqueous solution and could be stored for more than 10 days without any change in morphology. This will contribute to the synthesis of various branched metal NPs, thus promoting the application of SERS detection and efficient catalytic technology.

### 2.2. Preparation of Sea Urchin-like Composite Micro-Nanoparticles

In order to further functionalize sea urchin like micro-/nanoparticles, so as to improve the properties of materials urchin-like composite micro-/nanoparticles are often formed by coating the particles or mixing the guest into the matrix.

#### 2.2.1. Coated Urchin-like Nanoparticles

At present, there are two types of coated urchin-like composite micro-/nanoparticles. As shown in Figure 4a–c, the surface of urchin-like nano spikes is coated and the core-shell structure is formed by filling and cladding between the spikes as shown in Figure 4d. 

Liu et al. [67] developed a black urchin-like TiO_2–X_/Ag_3_PO_4_ particle. The synthesis steps are shown in Figure 4a: Firstly, sea urchin-shaped TiO_2_ was synthesized by a two-step hydrothermal method that was slightly modified. Then, the urchin-like TiO_2–X_ was generated under the action of NaBH_4_. Finally, urchin-like TiO_2–X_/Ag_3_PO_4_ particles were synthesized by in-situ reduction. SEM showed that the urchin-like structure was composed of a large number of nanospikes, and Ag_3_PO_4_ nanoparticles were evenly distributed on the surface of the nanospikes. Moreover, the number of Ag_3_PO_4_ nanoparticles increased with the increase in Ag^+^ concentration.

Wang et al. [68] developed urchin-like MgCo_2_O_4_@ppy core-shell composite by combining hydrothermal method and in-situ chemical oxidative polymerization (Figure 4b). The result showed that the urchin-like MgCo_2_O_4_ grown on NF provided a large number of active sites for chemical reactions due to its high specific surface area. At the same time, ppy in the outermost layer of the particle provided a transverse channel for electron transfer and thus reduced the internal resistance. The perfect combination of the two properties provided good physical and chemical conditions for ion diffusion and rapid electron transfer.

Jiang et al. [69] prepared a ZnO/Au/G-C_3_N_4_ composite heterostructure with an urchin-like micro-/nanostructure. The preparation method is shown in Figure 4c: (I) three-dimensional urchin-like ZnO with nanorods was prepared by a hydrothermal method with ZnO seed as the core material; (II) a HAuCl_4_ precursor was directly reduced to Au NPs by ultrasonic treatment on urchin-like ZnO nanorods in an ethanol–water blended system; (III) the g-C_3_N_4_ was nucleated in a crucible by a thermal vapor condensation (TVC) method and deposited on 3D urchin-like ZnO/Au. The composite particle size was about 5 µm. The diameter range of nanorods was 200–300 nm, and the size of Au NPs was 30–60 nm, as shown in Figure 4c.

Zhou et al. [71] prepared ZnO with TiO_2_ thin layer by a simple hydrothermal method. The diameter of the urchin-like composite was 5 µm with a good coating effect of TiO_2_ and a regular morphological structure. The influence of the pH value of the solution on the morphology of the urchin-like ZnO and the effect of the Ti(SO_4_)_2_ mass ratio, reaction time, and reaction temperature on the photocatalytic activity of the composite catalyst were investigated.

Chen et al. [70] reported a method to prepare urchin-like core-shell carbon spheres (SUCSs) by a one-pot cooperative assembly process as shown in Figure 4d. Tetraethyl orthosilicate(TEOS), 3-aminophenol/formaldehyde(AF), and cetyltrimethylammonium bromide (CTAB) were used as raw materials. In the alcohol–water system, the formation rate of AF resin was faster than the hydrolysis rate of TEOS under ammonia catalysis, and the resultant AF resin particles were nucleated. As the reaction proceeded, the concentration of 3-aminophenol decreased, and the formation rate of AF resin matched the hydrolysis rate of TEOS. AF resin polymer and silicate oligomer were assembled with CTAB through electrostatic interaction to form SiO_2_@AF resin polymer. Finally, after the silicon carbonized and etched, carbon spheres with a sea urchin-like core-shell could be obtained. The diameter and shell thickness of core-shell carbon spheres could be adjusted by adjusting the amount of TEOS and resorcinol formaldehyde resins. It was found that the structure had highly connected pore channels and accessible transport paths, which not only provided a large number of electrically active sites for easy access to the dielectric, but also benefited the rapid electron transition and high rate of electrolyte infiltration.

#### 2.2.2. Decorative Urchin-like Nanoparticles

It has been proved that combinations with other materials introduce different interfaces, leading to deterioration of the cyclic stability of electrode materials. To avoid the introduction of different interfaces, a straightforward strategy is to prepare homogeneous materials by decorating. According to different reaction mechanisms, decorating can be divided into two types. One is to directly form the decorated heterostructure nanorods and self-assemble them into urchin-like micro-/nanoparticles, as shown in Figure 5a,b; the other is to react to generate urchin-like micro-/nanoparticles and then conduct in-situ reduction decorating on the surface, as shown in Figure 5c,d.

Tada et al. [72] synthesized TiO_2_-SnO_2_ nanorods by a high-temperature method in the presence of SnO_2_ seeds with self-assembly into urchin-like structures, as shown in Figure 5a: Firstly, Ti(OBu)_4_ and SnO_2_ seeds were hydrolyzed in HCl solution. After 1 h at high temperature, rutile TiO_2_ appeared on the surface of SnO_2_. The adsorption of Cl^−^ on TiO_2_ plane restricted their growth to induce anisotropic growth along the direction of (001), forming TiO_2_-SnO_2_ nanorods. Finally, in the strong acid environment, due to the difference of zero charge between TiO_2_ and SnO_2_, the surface positive charge of SnO_2_ on the head was smaller than that of TiO_2_ on the tail. In addition, the van der Waals attraction of SnO_2_ particles was larger than that of TiO_2_ particles. Therefore, the head of one-dimensional nanoparticles had a small repulsive force and a large attractive force, resulting in the formation of three-dimensional radial urchin-like microspheres with their heads oriented toward the center.

Liu et al. [73] synthesized urchin-like NiO-NiCo_2_O_4_ microspheres with a heterostructure by using a two-step synthesis method as shown in Figure 5b. Firstly, the precursor of NiO-NiCo_2_O_4_ was prepared by hydrothermal reaction. Then, NiO-NiCo_2_O_4_ heterostructure microspheres were obtained by calcination of the precursor. The size of urchin-like microspheres was 5 µm in diameter, with a large number of small nanorods growing radially from the center. It was found that compared with mixed decorating or uniform decorating, the heterostructure formed by the coupling of nanocrystals with different band gaps had an unprecedented interface effect, which could promote the rapid transmission of ions and electrons, so as to improve the performance of the battery.

Ye et al. [74] reported that WO_3_/Au nanoparticles were prepared by in situ redox reaction with oxidizing metal salt precursors on urchin-like WO_2.72_ nanoparticles that have weak reductive properties in aqueous solution, as shown in Figure 5c. This synthetic strategy had the advantages that it took place in one step and required no foreign reducing agents, stabilizing agents, or pretreatment of the precursors, so as to avoid the introduction of impurities and ensure the clean interface of metal WO_3_ and uniform metal dispersion. The particle size distribution was narrower than 1 nm and adjustable. The hybrid layered structure was composed of a large number of radial nanowires with a diameter of 5–15 nm and a length of 600 nm.

Zhang et al. [75] combined the urchin-like Bi_2_S_3_ with good visible light absorption performance and Ag nanoparticles with a localized surface plasmon resonance effect to build a Bi_2_S_3_/Ag nano-heterostructure, as shown in Figure 5d. It found that with the introduction of Ag nanoparticles, the urchin-like structure on its surface was not damaged. The urchin-like Bi_2_S_3_/Ag was self-assembled from a single nanorod with a diameter of about 20 nm and a length of about 800 nm. Ag particles were 5 nm in diameter and distributed on the surface of the nanoparticles. Figure 5d(I–IV) clearly shows the Bi, Ag, and S elements were evenly distributed in the structure of the sample. It was found that the incorporation of Ag into urchin-like Bi_2_S_3_ not only produced a deeper effect, improving the light absorption capacity and photothermal conversion efficiency in the near-infrared region, but also accelerated the separation capacity of photogenerated carriers, thus improving the photocatalytic efficiency.

Dai et al. [76] used KMnO_4_, Al_2_ (SO_4_)_3_·18H_2_O and MnSO_4_·H_2_O as raw materials to synthesize urchin-like Al-decorated MnO_2_ supercapacitor materials in one step by a hydrothermal method. The morphology of the sample was an urchin-like microsphere, which was composed of nanowires with a diameter of about 30 nm. The material characterization showed that Mn, Al, and O elements were evenly distributed in the structure of the sample.

### 2.3. Preparation of Sea Urchin-Shaped Hollow Micro-/Nanoparticles

Hollow nanostructures have many potential applications in catalysis, drug delivery, optical imaging, and nanoreactors because of their large specific surface area, low density, good surface permeability, and strong loading capacity [77,78]. Combined with the characteristics and advantages of urchin-like nanostructures, three-dimensional hierarchical urchin-like hollow nanostructures have attracted extensive attention.

At present, the preparation of urchin-like hollow microspheres is mainly divided into the following methods according to the principle (1) the metal oxide is transformed into an urchin-like hollow structure in solution by using the self-assembly process based on the modified Kirkendall effect [79]; (2) the metal powder is thermally evaporated through the vapor–liquid–solid growth mechanism at a relatively high temperature to form the urchin-like hollow structure [80,81]; (3) nanomaterials are deposited on different templates to obtain core/shell composite structures, and then the templates are selectively removed by chemical etching or thermal decomposition to form urchin-like hollow structures [82,83]. According to the different growth processes of the urchin-like structure, it can be divided into a step-by-step method and a synchronous method. The process of the step-by-step method is to form a spherical structure and then generate nanorods on the surface of the sphere, as shown in Figure 6a–d, while the synchronous method is to generate the spherical structure and nanorods at the same time, as shown in Figure 7a–d. In the presence of the template, the template can be removed by calcination or dissolution to form urchin-like hollow microspheres. 

#### 2.3.1. Step-by-Step Method

In the step-by-step method, the growth of nanorods on the surface of the sphere is mainly obtained by the epitaxial growth of small nanocrystals on the shell layer of the sphere structure along a certain direction. The hollow structure can be formed in the process of forming the sphere or transformed into a hollow structure after the formation of the urchin-like sphere. At present, the hollow structure is mostly obtained by using a template.

Elias et al. [82] used modified polystyrene microspheres as a template and deposited ZnO nanocrystals on the surface of PS microspheres by an electrochemical method. As the reaction prgressed, ZnO nanorods grew along the direction of (0001) and formed homogeneous and stable urchin-like spheres. Cheng et al. [83] carbonized glucose into C as the template and zinc nitrate as the zinc source. Under the action of urea, Zn^2+^ was hydrolyzed by simple hydrothermal to form ZnO particles and was adsorbed on the C surface. With the progress of the reaction, ZnO formed nanorods along the (0001) direction. Finally, after calcination in air at 500 °C and 600 °C to remove the carbon core, urchin-like ZnO hollow spheres were obtained.

Due to the high specific surface energy of the ZnO polar surface, it is easy to grow preferentially along the (0001) direction to form nanorods. Therefore, there are many studies on urchin-like ZnO hollow microspheres, but there are also a few reports on the preparation of urchin-like hollow structures from PANI, TiO_2_, and Fe_3_O_4_.

Wang et al. [85] developed a new interfacial polymerization method, which diffuses aniline from a hollow spherical cavity through a hydrophilic channel for oxidative polymerization. The diffused aniline salt was dispersed and reacted with ferric chloride. PANI nanofibers were formed and assembled into urchin-like polystyrene/PANI hollow microspheres on the surface of hollow polymer spheres in aqueous solution. The diameter of PANI nanofibers was 15–20 nm and the length was 50–180 nm. The schematic illustration of the formation of the urchin-like PANI hollow microsphere is given in Figure 6b. Commercial PS hollow spheres containing a hydrophilic thin inner-layer and transverse channel were immersed in the anilinium salt solution, which permeated into the cavities of the PS hollow spheres through the hydrophilic channels. Then, they were dispersed into the ferric chloride solution, anilinium salt slowly permeated out from the transverse channel and reacted with ferric chloride, resulting in the formation of nanofibers on the external surface of the PS template, forming a unique urchin-like structure. Finally, the hollow template was removed by tetrahydrofuran treatment. It is noteworthy that the process uses hollow polymer spheres as template and carrier and simultaneously carries out the growth of PANI fibers and the assembly of complex organic structures. The monomer and oxidizer are separated by the polymer shell. Their diffusion makes the reactants gather together and oxidation polymerization occurs near the hollow shell. Thus, the template shell not only provides an interface for separating the reactants, but also provides a nucleation site for growing the fibers. The method is simple and easy to operate. However, an expensive polystyrene hollow sphere template and ferric chloride oxidant are used.

Tai et al. [92] used sulfonated PS microspheres that could be prepared in common laboratories as the template and cheap ammonium persulfate as the oxidant to prepare perfect urchin-like PANI microspheres with a diameter of 1.5 µm. The formation mechanism was investigated by time-dependent experiments. The sulphonation treatment enhanced the surface polarity of PS microspheres, which was beneficial for the growing of the aniline monomer onto the surface of PS microspheres, and a thin PANI shell was formed. With the increase in thickness, the short polyaniline nanofibers grew and covered the surface of the SPS microspheres. As the reaction progressed, the content of oxidizer decreased gradually, and the formed polyaniline nanofibers extended outward to capture more oxidizer to react with the monomer. Consequently, urchin-like PANI microspheres were obtained.

Zhou et al. [84] selected SiO_2_ spheres as templates and prepared urchin-like hollow TiO_2_/Ag particles in a step-by-step method. The synthesis process is shown in Figure 6a. Colloidal SiO_2_ nanospheres were first synthesized via a modified Stöber method, Then, the TiO_2_ shell was coated on the surface of SiO_2_ nanospheres by a sol–gel method in which tetrabutyl titanate (TBOT) was hydrolyzed and condensed in ethanol/ammonia alkaline solution. After SiO_2_@TiO_2_ core-shell nanospheres were treated in NaOH aqueous solution at 120 °C for 2 h, a novel urchin-like yolk–shell structure with radial extension of nanofibers was obtained. Then, highly crystalline urchin-like TiO_2_ hollow microspheres were obtained by acid treatment and calcination. Finally, Ag nanoparticles were attached to the nanofibers of the nanoparticles via in situ reduction of AgNO_3_ with trisodium citrate.

In the above method, irrespective of the template used to prepare urchin-like hollow micro-/nano spheres, the hollow size of the microspheres can be easily controlled through the design of template size. However, the use of hard/soft templates and surfactants or polymers also has some disadvantages. This requires high pressure, high temperature, or a complex process. In the process of removing templates, the shell may be damaged or collapsed, and impurities may be introduced, which may lead to blockage and adhesion to the surface of the active area of micro-/nano materials. At the same time, the template surface usually needs to be modified in the preparation process, which leads to more complex process steps. In order to simplify the process and avoid the adverse effects caused by the existence of formwork, Hen et al. [80] used zinc powder as raw material to prepare urchin-like ZnO hollow microspheres by thermal evaporation under atmospheric pressure. This process does not use any additives, which can effectively avoid introducing impurities to contaminate crystals. The formation mechanism is that Zn powder is heated in the high-temperature area to form Zn steam, which is transferred to the low-temperature area by argon as a carrier gas. Zn steam is rapidly condensed to form droplets, deposited on the silicon substrate, and solidified rapidly to form Zn balls. The surface of Zn balls is oxidized by the residual O_2_ in the system to form a thin layer of ZnO, and ZnO is continuously formed as the internal Zn is further evaporated and migrates to the surface. The ZnO grows directionally on the shell surface to form small ZnO nanorods. With the depletion of Zn, urchin-like ZnO hollow microspheres are finally obtained.

Bao et al. [86] reported that hollow urchin-like ZnO microspheres were prepared by two-step hydrothermal growth, which was a facile, low-temperature method, and no template was required. The size of the urchin-like ZnO microspheres was approximately 1.8 µm with an average shell thickness of 400 nm, and the ZnO nanorods were 400 nm long with a diameter of 100–200 nm. The synthesis steps are shown in Figure 6c. The growth mechanism was mainly to use the inhibition of sodium citrate to form ZnO hollow microspheres in the primary hydrothermal step, and then driven by the interfacial reaction between Zn^2+^ cations and OH^−^ anions, ZnO nanorods were grown readily around the surface of ZnO hollow microspheres to form an urchin-like structure in the second hydrothermal step. The hollow size and nanorods size can be easily controlled by adjusting the annealing temperature in the first step and the reaction time in the second step.

Cao et al. [87] synthesized novel urchin-like and yolk–shell TiO_2_ microspheres by a two-step method without using any template. As shown in Figure 6d, uniform TiO_2_ microspheres were firstly prepared by the sol–gel method, and the their monodisperse properties were finely regulated by surfactants such as KCl and aniline. Then, the NaOH-assisted hydrothermal method was used to further obtain the yolk–shell urchin-like structure. The characterization showed that the microsphere had a clear egg yolk shell structure, and the shell was formed by many urchin-like burrs, which were distributed on the surface of the microsphere. The authors found that the diameter and shell thickness of the UYTMs could be largely controlled by the concentration of NaOH, and KCl played a key role in inhibiting particle aggregation and promoting the production of highly spherical particles. The special hollow yolk–shell structure had a void inside, which increased the refraction of light and was conducive to improving the catalytic efficiency of light. In recent years, some research works have been done to simplify the process of preparing urchin-like hollow microspheres by using a step-by-step method, but the process is still cumbersome, and there are too many operating steps. Therefore, the purity and yield of the product need to be further improved.

#### 2.3.2. Synchronizing Method

Synchronous method is mostly obtained through Ostwald ripening process. Compared with the step-by-step method, the template used in the preparation of urchin-like hollow microspheres by synchronous method is more all kinds of bubbles. Zhou et al. [93] used the H_2_ bubble generated by NaBH_4_ as a template to induce the growth of ZnO nanocrystals on its surface. With the increase of temperature in the autoclave, the H_2_ bubble gradually disappeared with the increase of pressure, thus forming a hollow structure. Du et al. [94] took the CO and CO_2_ bubbles generated by the decomposition of H_2_C_2_O_4_ as the template, and the Fe_2_O_3_ nanocrystals generated by the reaction of Fe^3+^ compound and oxalic acid gathered on the bubble surface. Through the Ostwald ripening process, the Fe_2_O_3_ crystals inside the sphere slowly dissolved, and the nanocrystals grew into Fe_2_O_3_ nanorods along the epitaxial direction, so as to obtain the urchin-like Fe_2_O_3_ hollow structure. Compared with the hard template method, using various bubbles as the template can effectively solve the problems caused by impurities and template defects. However, the bubble size formed in the reaction process is uneven, resulting in uneven particle size of urchin-like hollow microspheres. At the same time, because the bubbles formed in the reaction process are easy to rupture, the morphology of sea urchin-like hollow microspheres is inconsistent. In order to overcome this problem, Li et al. [88] prepared sea urchin-like a-MnO_2_ hollow microspheres by Ostwald ripening mechanism without using any template. The reaction mechanism is shown in Figure 7a. They reacted KMnO_4_ acidic solution with Cu foil to form a-MnO_2_ colloid and aggregated into microspheres. Due to the one-dimensional growth habit of a-MnO_2_ crystal, the nanorods grew epitaxially from the surface of the initial colloidal microspheres along the (001) direction, forming an urchin-like structure. With the extension of reaction time, the internal MnO_2_ crystal nucleus gradually dissolved because it had higher surface energy. It grew on the external nanorods through Ostwald ripening mechanism until the internal crystal nucleus was completely consumed, and finally formed urchin like a-MnO_2_ hollow microspheres.

Cheng et al. [95] synthesized TiO_2_ hollow microspheres with an urchin-like layered structure by a simple one-step hydrothermal method; the size was 3 µm. The authors believed this occurred because potassium titanium oxide oxalate has an octahedral structure in which each Ti atom is coordinated by two [C_2_O_4_]^2−^ ions and bridged by two O atoms. Rutile TiO_2_ crystals were formed by decomposition in H_2_O_2_ solution. Finally, urchin-like hollow spheres were obtained by self-assembly of O_2_ bubbles decomposed with H_2_O_2_ as a soft template. The authors also studied the ER behavior of urchin-like TiO_2_ hollow microspheres under steady and oscillatory shear. Under the same electric field strength, hollow TiO_2_ suspension had a higher yield stress than that of pure TiO_2_ suspension. At the same time, the urchin-like structure made the hollow TiO_2_ suspension have stronger interfacial polarization and higher ER activity under the action of electric field. In addition, the hollow inner layer increased the long-term stability of the suspension and further improved the ER effect.

Wang et al. [89] prepared urchin-like hollow gold particles with Ag nanoparticles as a template and ascorbic acid as a reducing agent. The average particle size was 104 nm, and the internal particle size range was 23–45 nm. The reaction mechanism is shown in Figure 7b. The replacement reaction occurred when Ag nanoparticles were placed in HAuCl_4_ solution, resulting in the dissolution and migration of Ag atoms to the peripheral solution. HAuCl_4_ was reduced to Au atoms and they deposited on the original Ag surface. Along with the disappearance of Ag nanoparticles, Au nanoparticles shrank inward to minimize the surface energy. Since Ag diffused faster than the shrinkage of gold atoms, the shrinkage process stopped after Ag atoms were completely dissolved, leaving the central position unoccupied, so as to form a hollow structure.

Meng et al. [96] prepared hierarchical urchin-like γ-Al_2_O_3_ hollow microspheres by the hydrothermal method followed by a calcination process. With an average diameter of 2.5 µm, the microspheres were composed of a large number of well-arranged nanowires. The center of the urchin-like hollow microspheres was formed by interlacing and connecting pores of different sizes and shapes. The authors proposed that (PEO)_20_–(PPO)_70_–(PEO)_20_ block copolymer (P123), as a structural guiding agent, had dual functions in controlling the structure morphology, including adjusting the oriented attachment of nanowires and stabilizing the three-dimensional structure. In addition, acidic conditions may affect the oriented attachment of P123, which was conducive to the formation of three-dimensional urchin-like hollow superstructure.

Lim et al. [90] developed a simple template-free method to prepare urchin-like hollow Co_3_O_4_ precursors by the hydrothermal method and a heat treatment process (Figure 7c). From the image of the incomplete sphere in the Figure 7c(III), it can be seen that the core of this urchin-like precursor is hollow with typically 1–2 µm in diameter, and the nanowires are needle-like with a width of 200–300 nm at the front, about 100 nm in the middle, and 20–50 nm at the top. It shows that the formation process has the characteristics of lateral growth. The authors also studied the growth process of hollow sea urchin precursors and proposed a reasonable growth mechanism (Figure 7c). In the initial stage, small nanoparticles spontaneously aggregated into large spheres to minimize the overall surface energy of the system. Then, due to the one-dimensional growth tendency of nanorods, short nanorods were grown on their surface. As the reaction proceeded further, the inner core was dissolved, and Ostwald ripening helped the short nanorods grow into large needle-like nanowires.

Yang et al. [97] synthesized hierarchical urchin-like V_2_O_5_ hollow spheres by a simple and low-cost one pot hydrothermal method under the same reaction mechanism. The core diameter was 670–730 nm, and the urchin shape was composed of highly dense radial nanorods from the center to the outside, with a length of 1–1.3 µm. The diameter was about 50–70 nm.

Zeng et al. [91] developed a simple method to prepare and regulate hollow Fe_3_O_4_ @PDA- Ag microspheres with an urchin-like shape. The preparation route of microspheres is shown in Figure 7d. A facile micro-emulsion method was first used to produce urchin-like β-FeOOH hollow microspheres by using FeCl_3_·6H_2_O as an iron source and sodium dodecyl sulfate (SDS) as a morphology modifying agent. Then, the hollow microspheres were calcined in nitrogen atmosphere to obtain urchin-like Fe_3_O_4_ hollow microspheres. The PDA layer was coated on the surface of Fe_3_O_4_ particles at room temperature to obtain Fe_3_O_4_@PDA microspheres. Finally, Fe_3_O_4_@PDA microspheres were treated with ammonia aqueous solution at room temperature, and Ag nanoparticles were grown in situ on the surface of Fe_3_O_4_@PDA to form Fe_3_O_4_@PDA-Ag microspheres. In the sample preparation process, PDA was not only used as reducing agent, but also used as the supporting template of silver nanoparticles, which also helped to improve the catalytic and adsorption capacity in the process of dye removal.

Hu et al. [98] produced urchin-like MoG@NiCo-precursor@C hollow microspheres by using glycerol molybdenum nanosphere as a template, with further transformation to MoS_2_/NiCo_2_S_4_@C HMSs by a sulfidation reaction of thioacetamide in ethanol. It is noteworthy that MoG NSs and glucose in the reaction process had an important influence on the morphology of high-altitude hole hybrid microspheres. As a template, MoG NSs provided a center for the growth of NiCo precursor needles, and a thin carbon layer obtained from glucose could maintain the pristine sphere morphology. The microspheres had the advantages of synergistic binary metal composition, short diffusion paths, abundant active surfaces, the ultrahigh rate capability. It was found that the thin carbon layer could prevent the aggregation and dissolution of active components and enhance the cycling stability.

Fan et al. [99] prepared urchin-like hollow Co_3_O_4_ microspheres by a two-step hydrothermal method and post calcination industry. In the first step, cobalt alkanol microspheres were prepared by the hydrothermal method. The second step was conversion into hollow urchin-like COOH by the hydrothermal method. Finally, urchin-like hollow Co_3_O_4_ microspheres were obtained by calcination. The results showed that the surface of the microspheres had a unique hollow structure and mesoporous nanosheets. The average diameter of microspheres was 1 µm, and they were composed of ultra-thin nanosheets with an average surface thickness of about 5 nm. There were 2–5 nm holes on the nanosheets, and the hollow inner diameter was about 0.8 µm.

## 3. Applications

### 3.1. Photocatalyst

Urchin-like microspheres are considered promising photocatalytic materials because of their large specific surface area and significantly improved catalytic activity compared with existing photocatalysts. The catalytic mechanism is that under UV irradiation, it is excited to produce photogenerated electron hole pairs to form highly active groups with strong redox ability such as −OH and −O_2_, so as to react with organic pollutants and degrade them into small molecular substances. Due to the spherical surface and one-dimensional nanorods, urchin-like microspheres can provide more surface-reaction points and light-scattering points. Therefore, compared with conventional materials, urchin-like microspheres can produce more photogenerated electron hole pairs, form more high-performance groups such as −OH and −O_2_, and have stronger photocatalytic activity.

Xu et al. [100] prepared urchin-like TiO_2_ by a template-free solvothermal method using acetone as the solvent and hydrochloric acid as the morphology control agent. Under UV with the wavelength centered at 365 nm irradiation, the photocatalytic degradation activities of the sample and P-25 for phenol were compared. The authors suggested that the high catalytic activity of the sample was due to the layered urchin-like structure, which expanded the specific surface area (134.6 m^2^/g) of the sample, enhanced the light capture ability, the fast electron transfer rate, and the good crystallinity. Liu et al. [101] also found that urchin-like anatase TiO_2_ had a high degradation rate of methyl orange and phenol.

Since the performance of single component urchin-like microspheres is limited, double shell urchin-like microspheres can exert the degradation of multiple components. Tada et al. [72] developed a radial heteromesocrystal photocatalyst composed of SnO_2_ (head) and rutile TiO_2_ nanorods (tail), which had significant photocatalytic activity for partial oxidation of ethyl alcohol. The authors believed that because the TiO_2_ nanorods that were several microns long could effectively absorb light, the electrons in the conduction band of TiO_2_ were transported in the (001) direction with high conductivity and were effectively transferred to the SnO_2_ through the high-quality interface. The heteroepitaxial junction-induced CB-band bending in SnO_2_ enhanced charge separation.

Zhao et al. [102] prepared sea urchin-like rutile TiO_2_ modified by Au or Ag nanoparticles by a wet chemical method, with a size of about 1–3 µm and a specific surface area of 40 m^2^/g. The photocatalytic experiment showed that the photocatalytic activity of TiO_2_ modified with noble metals for methyl blue in water was enhanced. This was because the interfacial electron transfer between TiO_2_ and metal nanoparticles enhanced charge separation. The authors proposed that under light irradiation, the surface plasmon resonance of noble metal nanoparticles could not only enhance the light trapping ability but also induce photoexcited electrons in the noble metal nanoparticles on TiO_2_ to enter the conductive band (CB) of TiO_2_ through the noble metal-TiO_2_ interface. The electrons in the CB of TiO_2_ could produce superoxide radicals, which can be used for the degradation of organic dyes as shown in Figure 8a. In addition, 3D nanostructures were easier to separate from solution than 0D or 1D nanostructures.

Wei et al. [105] studied the photocatalytic degradation of methyl orange by urchin-like In_2_O_3_/ZnO hollow composite microspheres under visible light and compared them with pure ZnO and In_2_O_3_ particles. It was found that pure ZnO showed poor degradation ability due to its large energy band, and pure In_2_O_3_ showed certain photocatalytic activity. After 4-h irradiation under visible light, the degradation efficiency of methyl orange was 65%, while the photocatalytic degradation efficiency of urchin-like In_2_O_3_/ZnO hollow composite microspheres reached 79% and had good stability.

Jin et al. [91] found that compared with smooth spheres, the catalytic reduction efficiency and adsorption capacity of urchin-like Fe_3_O_4_@PDA-Ag hollow microspheres for organic dyes (methylene blue and rhodamine) were significantly enhanced under different pH conditions. The authors attributed this to the urchin-like hollow structure and chemical composition of the sample in which the immobilized Ag nanoparticles had a synergistic effect with the PDA layer and sea urchin-like Fe_3_O_4_ hollow core. Moreover, the microspheres had good magnetism, high reusability, easy separation, and rapid regeneration ability. After recycling, the catalytic efficiency and adsorption efficiency did not decrease significantly.

### 3.2. Electrochemical

In energy applications, the surface area is a major feature of electrode materials because it significantly affects charge and discharge rates, which are among the main performance indicators of modern energy storage. Sea urchin-like micro-/nanoparticles have high porosity, which can mediate ion entry into the electrode, improving exchange dynamics and the overall efficiency of energy-related devices.

Jiang et al. [69] successfully prepared a three-dimensional urchin-like ZnO/Au/graphite-carbon nitride (g-C_3_N_4_) composite heterostructure, which as a photocathode had significantly enhanced photocurrent density and high far-field efficiency (Figure 8b). The vector study of Au nanoparticles showed that Au played a role in promoting the transfer of electrons. The 3D urchin-like micro-/nano structure inhibited the recombination of photogenerated electron holes, promoted the directional transport of carriers, and made it have excellent PEC properties. In addition, Pt NP-loaded three-dimensional urchin-like ZnO/Au/g-C_3_N_4_ photocathode showed an excellent hydrogen production rate (3.69 μmol h^−1^ cm^−2^) and a Faraday efficiency of 95.2%.

Chen et al. [106] reported a Li-S battery material composed of urchin-like cobalt nanoparticle embedded and nitrogen-doped carbon nanotube/nanopolyhedra (Co-NCNT/NP) superstructures. Because of the hierarchical micro-mesoporous structure in Co-NCNT/NP, it could effectively impregnate sulfur and physically limit and block the diffusion of soluble polysulfides. Moreover, connection to the Co-NCNT/NP conductive network could promote electron transport and dielectric penetration. Therefore, the battery provided a high discharge capacity and stable cycle performance.

Hydrogen is expected to be a candidate to replace traditional fossil dyes such as coal, oil, and natural gas because of its high energy, recyclability, and zero carbon emissions. Photoelectrochemical (PEC) water separation technology has great potential in large-scale conversion of solar energy into hydrogen [107], while 3D urchin-like micro-/nanospheres can be used in the design of high-performance PEC batteries because of their high specific surface area and strong light collection ability. Hao et al. [108] reported ternary ZnIn_2_S_4_ -Au-TiO_2_ Z-scheme heterostructure photocatalysts. The structure had a high specific surface area and wide light absorption range, which greatly improved the photocatalytic efficiency. In the test of connecting for 7 h, the obtained photocatalysts achieved the highest H_2_ production at the rate of 186.3 µmol g^−1^ h^−1^, and the O_2_ production rate reached 66.3 µmol g^−1^ h^−1^.

Sun et al. [109] electrodeposited porous urchin-like Ni_2_P microsphere on nickel foam (Ni_2_P/Ni/NF) as bifunctional electrocatalysts for overall water splitting. The optimal Ni_2_P/Ni/NF exhibited remarkable catalytic performance for both the HER and OER in an alkaline electrolyte (1.0 M KOH) and showed robust stability. The authors believed that the superior activity and strong stability of Ni_2_P/Ni/NF are closely related to its electrochemically active constituents, 3D interconnected porosity, high electrochemical surface area, and the close interaction between catalyst and conductive nickel foam.

Wang et al. [52] produced hierarchical urchin-like MnCo-selenide on nickel foam. The unique hierarchical microstructure, synergetic effect, and excellent conductivity enabled the electrode to exhibit outstanding supercapacitor performance compared with counterpart oxide and sulfide, including high specific capacitance (1656 F g^−1^ at 1 A g^−1^) and extraordinary cycle performance (8.2% capacity decline after 8000 cycles), as shown in Figure 8c. In addition, the authors also found that the asymmetric supercapacitor composed of manganese selenide and activated carbon had an energy density of 55.1 wh/kg at 880 w/kg, which was a satisfactory energy storage material.

Jiang et al. [43] prepared a 3D, highly conductive urchin-like NiCo_2_S_4_ nanostructure by a facile precursor transformation method. The synthesized sample had a lower optical band gap energy and higher conductivity, which provided more abundant redox reactions than two single component sulfides Co_9_S_8_ (Figure 8g(III)) and NiS (Figure 8g(IV)) or two corresponding single component oxides. In Figure 8g(I,II), there are two separate plateaus in the charge or discharge process, which demonstrate the dual redox processes of the active material while charging or discharging. The galvanostatic charge–discharge curves of the composite are highly symmetrical, and there is no obvious IR drop at low current density, indicating that the composite has fast I–V response and good electrochemical reversibility. It is worth noting that 91.4% capacitance can still be maintained in 5000 cycles, which is significantly higher than many binary sulfides NiS and CoS_2_ previously reported.

Zhang et al. [110] grew urchin-like ZnCo_2_O_4_ microspheres on nickel foam and used them as electrodes for the supercapacitor. The electrochemical performance of the electrode in an alkaline electrolyte was studied. When the current density was 1 A/g, the capacitance of the electrode was 1841.8 F/g and maintained the capacitance retention of about 78.4% at 10 A/g. The specific capacitance was about 1390.1 F/g after 3000 cycles, and the cycle stability was good (95.8%). The results showed that the sea urchin ZnCo_2_O_4_ microspheres supported on nickel foam had broad application prospects as high-performance electrodes for supercapacitors.

### 3.3. Other Applications

Cheng et al. [83] studied the changes of the complex permittivity and dielectric loss of urchin-like ZnO calcined at 500 °C versus frequency. It was found that the dielectric loss reached the maximum at 14.4 GHz (Figure 8d). The authors attributed it to the formation of oxygen vacancies at the interfaces of ZnO nanosheets; In addition, the sample also exhibited relatively higher EM wave absorption properties. As shown in the Figure 7d, the maximum RL can reach to −20 dB at 14.3 GHz. The electromagnetic wave absorption showed double peaks, and the weak absorption peak shifted towards lower frequency with increasing thickness. The peak shift was due to a quarter wavelength attenuation that occurred when the absorption met phase matching.

Yin et al. [103] prepared a novel electrorheological (ER) suspension composed of Cr-doped titania particles with a sea-urchin-like hierarchical morphology and demonstrated a distinct enhancement in ER properties. The study of ER properties showed that the suspension of graded Cr-doped titania had a stronger ER effect than that of smooth non-hierarchical Cr-doped titania suspension (Figure 8e). Furthermore, under electric and shearing fields, their rheological behavior was significantly different. The authors attributed it to the high specific surface area of nanostructures enhanced the interfacial polarization of particles; the high roughness and “interlocking” effect of unique urchin-like nanostructures significantly enhanced the friction and adhesion between particles. Based on the above results, it could be confirmed that the ER effect of titania could be improved not only by ion doping, but also by introducing mesoporous nanostructures with a high specific surface area into the particles. Therefore, the joint design of the chemical structure and mesoscopic structure of ER materials may provide a new way to obtain high ER effect.

Sun et al. [104] reported the selective and electric quantitative detection of formaldehyde on a gas sensor based on urchin-like In_2_O_3_ hollow spheres with a specific surface area up to 122.02 m^2^/g (Figure 8f(I)). The results showed that at the optimum working temperature of 140 °C, the sensor had the characteristics of sensitivity, selectivity, fast response, and a low detection limit. The sensor showed a good linear response in a relatively wide formaldehyde concentration range (50 ppb–10 ppm) (Figure 8f(II)), which showed its potential for quantitative detection of formaldehyde.

Sun et al. [38] prepared urchin-like In_2_O_3_ microspheres by the solvothermal method. On this basis, a new sensor was designed. The sensor showed excellent performance for O_3_ sensing at a low operating temperature, and the response was about 2.1–10 ppb at 150 °C. These results suggest that the urchin-like layered structure provided well-defined and well-arranged micropores and nanopores, which were conducive to the effective diffusion of gas. In addition, the large specific surface area provided rich active sites for gas chemisorption and reaction.

## 4. Conclusions and Outlook

### 4.1. Conclusions

In this paper, the preparation methods of urchin-like micro-/nanoparticles are described, with emphasis on hydrothermal/solvothermal methods, which are summarized in Table 1. Solvothermal and hydrothermal processes establish a very large reaction group, which is realized by increasing the pressure and temperature of the reaction in an autoclave. The composition, crystallinity, and shape of urchin-like particles formed during these processes are very sensitive to the type of precursor, solvent, pH of the solution, the reaction temperature, and reaction time. The various methods of urchin-like micro-/nanoparticles introduced above can follow different growth strategies, including Ostwald maturation, directional attachment, reaction-limited directional growth of nanocrystals of different crystal structures, seed growth triggered by self-assembly, surface metastable state phase, etc. Understanding the formation mechanism of nanoparticles is the key to controlling their morphology. The formation of nanoparticles can be described by the classical Ostwald maturation mechanism, in which the reduction of surface energy drives the growth of large particles at the expense of small particles. This phenomenon is mainly used to explain the preparation of thermodynamically stable nearly spherical nanocrystals. In wet chemical synthesis, seed nucleation and growth dynamics have a strong influence on the shape of nanostructures. The balance between the release rate of metal ions from solid matter and the coordination rate of metal ions to ligands is the key to control the growth of sea urchin-like particles. However, the understanding of the underlying growth dynamics and thermodynamics is lacking.

Selective exposure of desired crystal faces is crucial in the mechanism of selective directional growth. As discussed previously, the metal oxide ZnO [82] has preferential growth characteristics on the surface (1000), and γ-MnS [44], Co_3_O_4_ [90], and α-MnO_2_ [88] have a one-dimensional growth tendency, so no guidance is added. It should be noted that the reaction solvent has an important influence on the final morphology. However, the diversity of crystal structures of different materials, changes in the coordination environment of surface atoms, and common surface defects or vacancies complicate the reaction mechanism. This synthesis strategy is only suitable for some specific materials. In the case of controllable self-assembly of nanoparticle building blocks into more complex anisotropic nanostructures, organic coating reagents usually play a key role in reducing the surface activity of nanocrystals. Organic ligands with high affinity can selectively adsorb cap molecules (e.g., fatty acids, alkyl amines, amino acids, trioctyl phosphine oxide, block copolymers) on specific crystal faces of growing nanoparticles through surface adsorption and interaction between ligands to facilitate or regulate directed assembly. For example, the (100) plane of TiO_2_ is supported by F^−^ ions, while the form of Al_2_O_3_ is effectively controlled by sulfate ions. As mentioned above [72], Cl^−^ ions act as a habit modifier, and their preferential adsorption on the oxygen-defect sites on the rutile TiO_2_ (110) plane induces the anisotropic growth of TiO_2_ in the (001). However, the synthesis strategy is also applicable only to certain materials.

For materials without a one-dimensional growth tendency, reaction-decomposed gases can also be used to regulate and guide the growth of nanowires, such as ref. [93,94]. In the later introduction, hollow urchin-like micro-/nanoparticles were also prepared by using bubbles as templates, as seen in Table 2. Various bubbles used as templates can effectively solve the problems caused by impurities and template defects. However, the size of bubbles formed during the reaction process is uneven, resulting in uneven particle size of urchin-like inorganic hollow microspheres. At the same time, because the bubbles formed during the reaction rupture easily, the morphology of the urchin-like microsphere is often inconsistent.

The synthesis of materials with hierarchical nanostructures provides the opportunity to accurately adjust their physical and chemical properties to suit specific applications. In order to improve the functionality and potential applications of urchin-like micro-/nanoparticles, one possible strategy is to deposit urchin-like particles on a shape-controllable substrate to make electrodes, which are widely used in flexible supercapacitors, electrochemical splitting water, batteries, and other energy storage converters. This strategy has potential risks of substrate corrosion and contamination. To solve this problem, attempts can be made to develop techniques such as hot pressing, steam-assisted conversion, and solvent-free growth. Another common strategy is to compound it to achieve applications in other fields. For example, the above-mentioned ppy grown on MgCo_2_O_4_ provides transverse channels for electron transfer and reduces internal resistance. The perfect combination of the two properties provides good physical and chemical conditions for ion diffusion and rapid electron transfer. Alternatively, urchin-like Bi_2_S_3_ with good visible absorption properties can be combined with Ag NPs, which have an LSPR effect. This can not only produce a deeper effect, improve the photothermal conversion efficiency, and improve the light absorption capacity in the near infrared region, but also accelerate the separation ability of photogenerated carriers, so as to improve the photocatalytic efficiency. Table 3 summarizes the properties and applications of urchin-like composite micro-/nanoparticles. At present, the applications mainly focus on catalysis and electrochemistry, but there are few applications in other biomedicine, metamaterials, and smart materials. Urchin-like microparticles/nanoparticles have several properties that are useful for nanomedicine, for example, a good specific surface area, adjustable particle size, and functional surface. Metal incorporation can produce therapeutic effects, such as photothermal energy conversion. In addition, the combination with fluorescent materials facilitates diagnostic tasks such as imaging and sensing. Therefore, urchin-like micro-/nanoparticles can provide promising support for combining diagnostic and therapeutic tasks in one system.

### 4.2. Outlook

To sum up, the urchin-like micro-/nano structure has attracted extensive attention of researchers. Irrespective of the method, there are corresponding advantages and disadvantages, and the urchin-like structure parameters and morphological regularity are closely related to the preparation method, preparation conditions, and selected raw materials. Compared with solid materials, the photocatalytic degradation performance of urchin-like hollow microspheres has been greatly improved, which is expected to be widely used in the field of wastewater treatment. However, at present, research on urchin-like hollow microspheres is still relatively limited. In follow-up work, a lot of scientific researchers need to make numerous efforts to explore the basic scientific problems and promote their application. We believe that the development of urchin-like micro-/nano structures in the future will mainly include the following aspects:(1)The preparation method of urchin-like hollow microspheres is not advanced and is still in the stage of experimental exploration. There are few studies on the preparation of urchin-like inorganic hollow microspheres with itself as the template, which mostly requires a multi-step process, which is time-consuming and laborious. At the same time, the urchin-like hollow microspheres prepared by the existing methods usually have low morphology regularity, a large size, and poor stability. Therefore, developing new and more effective preparation methods to make the reaction conditions mild, controllable, environmentally friendly, and with a low cost, as well as make the prepared sea urchin-like inorganic hollow microspheres have a regular structure, small particle size, and uniformity is one of the main research directions in the future.(2)At present, the prepared urchin-like hollow microspheres have developed from single shell to a multi-shell structure, and the application performance of the materials has been greatly improved. However, most multi-shell urchin-like hollow composite microspheres are often prepared and play a role alone. How to obtain urchin-like hollow microspheres with a multi-shell at the same time is still facing challenges.(3)Because the urchin-like hollow microspheres have inner and outer surfaces and one-dimensional nanorods, they have a larger specific surface area and higher quantum yield than solid microspheres. However, at present, the research on the properties of urchin-like hollow microspheres is not in-depth, which has certain limitations for expanding their applications in various fields. Therefore, in-depth study on the correlation between structural parameters and properties of urchin-like hollow microspheres is significant for the development and application of urchin-like hollow microspheres.

## Figures and Tables

**Figure 1 materials-15-02846-f001:**
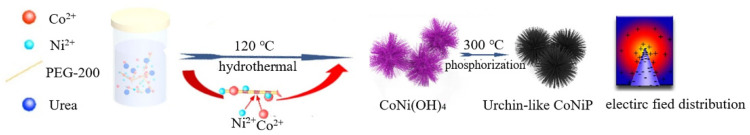
The self-assembly to form CoNi(OH)_4_ and urchin-like CoNiP in the presence of Ni^2+^ and Co^2+^ ions with the help of PEG [37] (Reprinted with permission from ref. [37]. Copyright 2021 IOP).

**Figure 2 materials-15-02846-f002:**
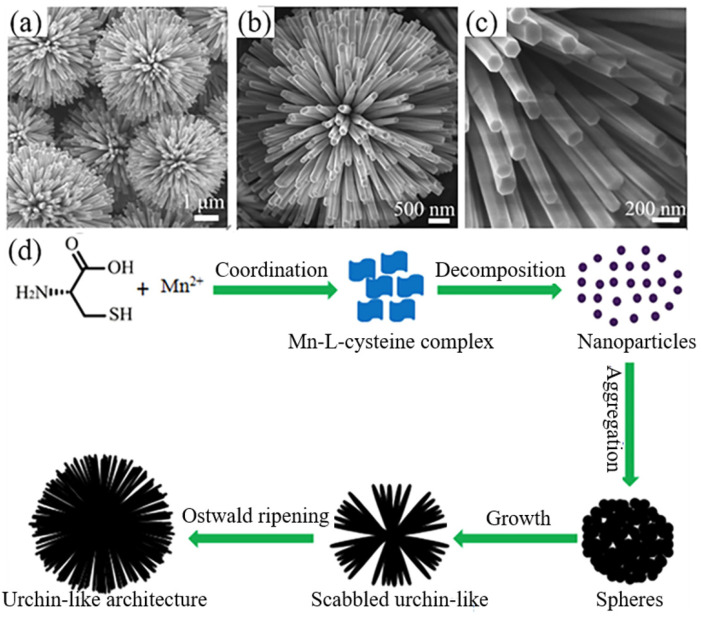
FESEM images of the as-prepared urchin-like γ-MnS architectures: (**a**) high-magnification SEM image, (**b**) enlarged SEM image of an individual urchin-like γ-MnS architecture, (**c**) FESEM image of a few 1D nanorods, (**d**) Schematic illustration of the growth mechanism of the urchin-like γ-MnS architectures [44] (Reprinted with permission from ref. [44]. Copyright 2019 Elsevier).

**Figure 3 materials-15-02846-f003:**
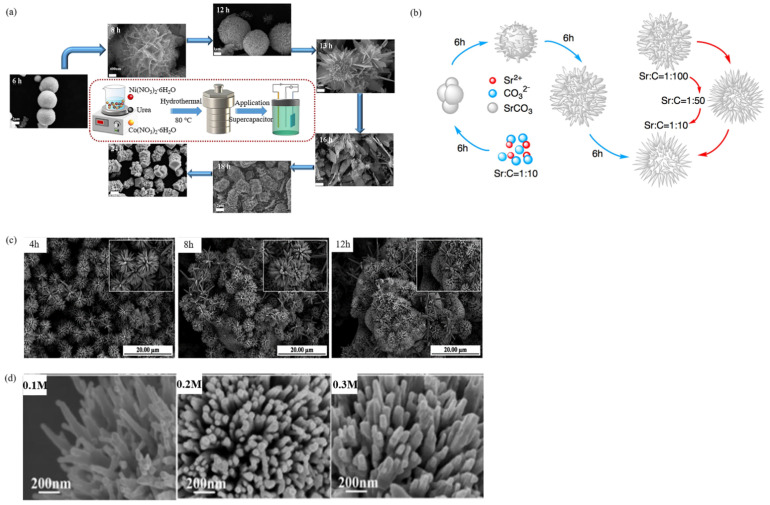
Effects of the reaction time and concentration of reactants on the morphology of urchin-like micro-/nanoparticles: (**a**) Preparation process of NiCo_2_O_4_ with different morphologies under different reaction times [47] (Reprinted with permission from ref. [47]. Copyright 2021 John Wiley & Sons Ltd.). (**b**) A schematic illustration of SrCO_3_ morphology evolution influenced by reaction time is in the left circle, that influenced by reactant concentrations is on the right [48] (Reprinted with permission from ref. [48]. Copyright 2017 Elsevier). (**c**) FESEM images of ZnO nanostructures prepared under different hydrothermal times [49] (Reprinted with permission from ref. [49]. Copyright 2016 Elsevier Ltd. and Techna Group S.r.l). (**d**) SEM images of products synthesized at different concentrations of the reaction solution [50] (Reprinted with permission from ref. [50]. Copyright 2019 World Scientific).

**Figure 4 materials-15-02846-f004:**
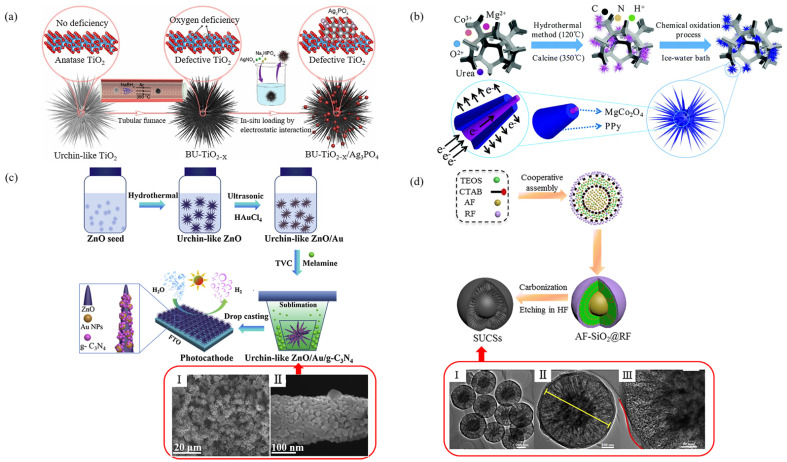
Examples of different composite types of sea urchin-like composite micro-/nanoparticles: (**a**) The synthesis process of BU-TiO_2–X_/Ag_3_PO_4_ [67] (Reprinted with permission from ref. [67]. Copyright 2021 Elsevier). (**b**) Schematic diagram of the synthesis process and electron transfer of MgCo_2_O_4_@PPy/NF [68] (Reprinted with permission from ref. [68]. Copyright 2018 The Royal Society of Chemistry). (**c**) Schematic diagram of the fabrication processes of the urchin-like ZnO/Au/g-C_3_N_4_ photocathode and the low-magnification (I) and high-magnification (II) SEM images of urchin-like ZnO/Au/g-C_3_N_4_ [69] (Reprinted with permission from ref. [69]. Copyright 2019 Elsevier). (**d**) Illustration of the fabrication of the SUCSs via one-pot cooperative assembly strategy and TEM images of SUCSs with different amount of RF resins; SUCSs-R-0.2 (I–III) [70] (Reprinted with permission from ref. [70]. Copyright 2017 Elsevier).

**Figure 5 materials-15-02846-f005:**
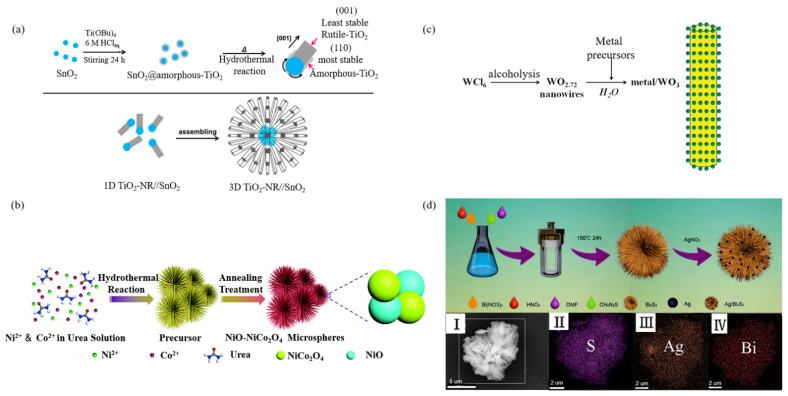
Examples of decorated urchin-like micro-/nanoparticles depending on reaction mechanism: (**a**) A schematic representation of the formation and self-assembly of TiO_2_ -NR//SnO_2_ [72] (Reprinted with permission from ref. [72]). (**b**) Schematic illustration of the formation of NiO–NiCo_2_O_4_ microspheres through a two-step method [73] (Reprinted with permission from ref. [73]. Copyright 2019 The Royal Society of Chemistry). (**c**) Schematic procedure for in situ loading of metal particles on WO_3_ [74] (Reprinted with permission from ref. [74]. Copyright 2012 American Chemical Society). (**d**) Schematic diagram of the preparation of the urchin-like Bi_2_S_3_/Ag nanostructures and the EDS spectrum of the Ag/Bi_2_S_3_ (I), and the elements of S, Ag, and Bi of the Ag/Bi_2_S_3_ (II–IV) [75] (Reprinted with permission from ref. [75]. Copyright 2021 American Chemical Society).

**Figure 6 materials-15-02846-f006:**
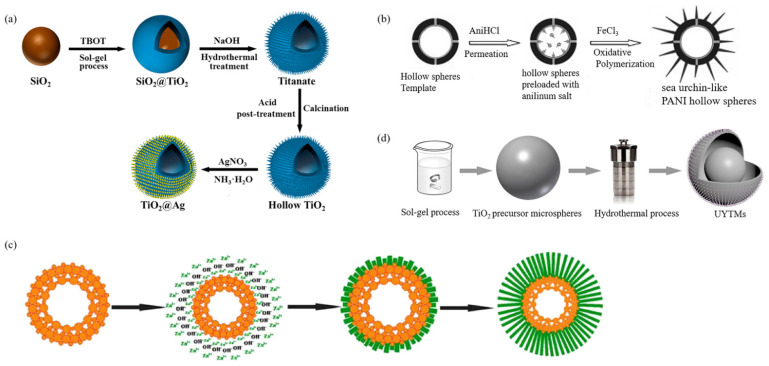
Examples of the preparation of urchin-like hollow micro-/nanoparticles by the step method: (**a**) Schematic illustration of the fabrication of hollow urchin-like TiO_2_@Ag NPs [84] (Reprinted with permission from ref. [84]. Copyright 2017 Elsevier). (**b**) Schematic representation of the formation of urchin-like polyaniline hollow spheres [85] (Reprinted with permission from ref. [85]. Copyright 2008 Elsevier). (**c**) Fabrication processes of hollow urchin-like ZnO microspheres [86] (Reprinted with permission from ref. [86]. Copyright 2016 Elsevier Ltd. and Techna Group S.r.l). (**d**) Schematic representation of UYTMs [87] (Reprinted with permission from ref. [87]. Copyright 2018 Taylor & Francis Online).

**Figure 7 materials-15-02846-f007:**
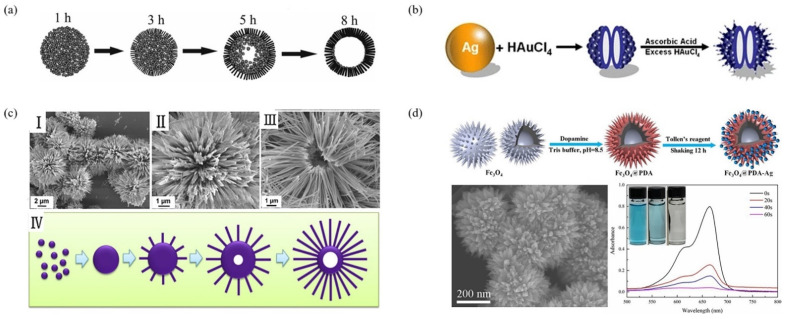
Examples of one-step preparation of urchin-shaped hollow micro-/nanoparticles: (**a**) Schematic Illustration of the Formation of R-MnO_2_ Hollow Urchins via the Ostwald Ripening Process [88] (Reprinted with permission from ref. [88]. Copyright 2006 American Chemical Society). (**b**) Scheme for the fabrication of hollow urchin-like AuNPs [89] (Reprinted with permission from ref. [89]. Copyright 2012 Springerlink). (**c**) Corresponding FESEM images at different magnifications (**I**,**II**) for the urchin-like Co_3_O_4_ spheres. (**III**) An incomplete sphere showing a hollow interior. (IV) Schematic illustration of the formation process of the urchin-like Co(CO_3_)_0.5_(OH)·0.11H_2_O hollow spheres [90] (Reprinted with permission from ref. [90]. Copyright 2012 Elsevier). (**d**) (I) Schematic of the synthesis route for urchin-like Fe_3_O_4_@PDA-Ag hollow microspheres, SEM images of (II) Fe_3_O_4_, (III) Time-dependent UV–vis spectra for adsorption of MB solution using urchin-like Fe_3_O_4_@PDA-Ag hollow microspheres as adsorbents [91] (Reprinted with permission from ref. [91]. Copyright 2018 Elsevier).

**Figure 8 materials-15-02846-f008:**
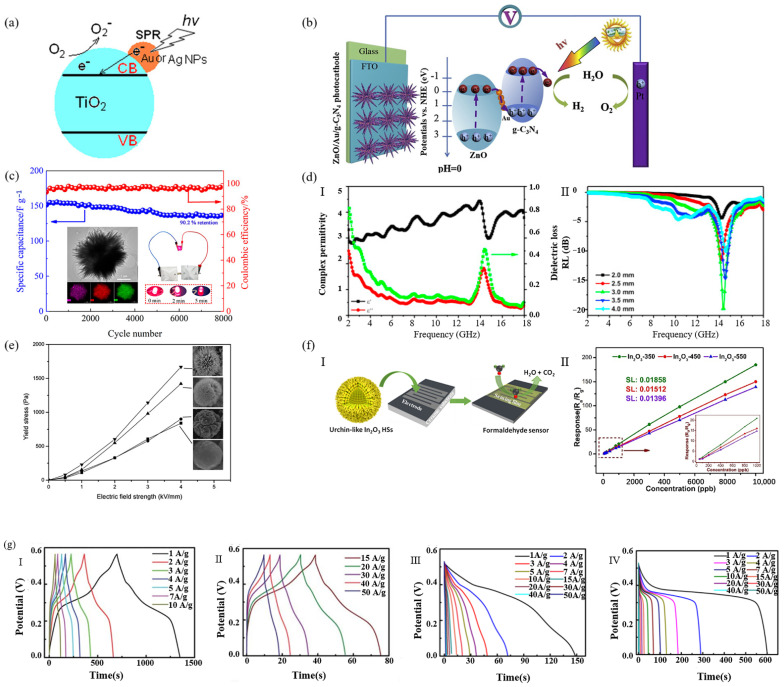
Overview of the application of urchin-like micro-/nanoparticles in different fields: (**a**) Schematic illustration of the surface plasmon-enhanced photocatalysis mechanism of Au or Ag-decorated TiO_2_ nanostructures under light irradiation [102] (Reprinted with permission from ref. [102]. Copyright 2013 Elsevier). (**b**) Schematic diagram of the PEC reaction mechanism of the 3D urchin-like ZnO/Au/g-C_3_N_4_ photocathode, EIS Nyquist plots [69] (Reprinted with permission from ref. [69]. Copyright 2019 Elsevier). (**c**) TEM of MnCo-selenide. TEM mapping of Co, Mn, Se. Images of the red LED lit by two ASCs in series [52] (Reprinted with permission from ref. [52]. Copyright 2019 American Chemical Society). (**d**) The complex permittivity (**I**) and RL curves (**II**) of calcined ZnO/paraffin wax composite [83] (Reprinted with permission from ref. [83]. Copyright 2015 Elsevier). (**e**) Yield stress as a function of electric field strengths for the suspensions of Cr-doped titania particles with different surface morphologies (T ¼ 23 °C, 10 vol%.) [103] (Reprinted with permission from ref. [103]. Copyright 2009 The Royal Society of Chemistry). (**f**) (**I**) Schematic illustration for sensor fabrication. (**II**) The linear relationship between the response and HCHO concentration (inset: 50–1000 ppb) [104] (Reprinted with permission from ref. [104]. Copyright 2019 Elsevier). (**g**) Electrochemical performances of the ternary NiCo_2_S_4_ urchin-like nano-structure: (**I**,**II**) galvanostatic charge–discharge curves measured with different current densities of 1–50 A g^−1^; (**III**) Co_9_S_8_; (**IV**) NiS [43] (Reprinted with permission from ref. [43]. Copyright 2013 The Royal Society of Chemistry).

**Table 1 materials-15-02846-t001:** Synthetic method and properties of solid urchin-like micro-/nanoparticles.

Method	Material	Solvent	Surfactant	BET Surface Area	Particle Size (Diameter)	Application	Ref.
hydrothermal	Co_3_O_4_	deionized water	CTAB	-	5–7 µm	lithium ion battery	[36]
		deionized water	-	165 m^2^/g	3 µm	energy storage	[51]
	CoNiP	deionized water	PEG-2000	-	2–5 µm	Hydrogen evolution reaction catalytic	[37]
	In_2_O_3_	ethanol	SDS	58.6 m^2^/g	1 µm	O_3_ gas sensor devices	[38]
	NiO	deionized water	-	-	1 µm	-	[41]
	CoP	deionized water	-	-	5 µm	electrocatalysts	[42]
	NiCo_2_S_4_	deionized water	-	20.33 m^2^/g	4 µm	high-rate supercapacitors	[43]
	γ-MnS	distilled water, EG	-	34.55 m^2^/g	4–5 µm	photocatalytic	[44]
	Bi_2_S_3_	deionized water	-	-	6–8 µm	-	[45]
		DMF, deionized water	-	-	2 µm	photocatalyst	[46]
	NiFeP	DMF		118.9 m^2^/g	5 µm	photocatalyst	[26]
	NiCo_2_O_4_	deionized water	-	158.6 m^2^/g	4 µm	supercapacitors	[47]
	SrCO_3_	deionized water	-	-	4 µm	capacitor	[48]
	α-Fe_2_O_3_	deionized water	glucose	151.2 m^2^/g	0.5–1.0 µm	-	[39]
	Fe_3_O_4_	deionized water	glucose	-	0.5–1.0 µm	microwave absorbing	[40]
	ZnO	deionized water	-	-	5–10 µm	Photocatalytic	[49]
	MnCo-selenide	deionized water	-	-	4.28 µm	capacitors	[52]
electrodeposition-hydrothermal	ZnMn_2_O_4_	ethanol and deionizer water	sodium 王citrate	25.34 m^2^/g	500 nm	electrodes	[53]
thermal decomposition	α-Fe_2_O_3_	deionized water	-	60.24 m^2^/g	400 nm–2.5 µm	electrodes	[54]
seeding growth approach	Au	deionized water	SDS	-	40 nm	-	[64]
hydrolyzing-heat-treating	Au	deionized water	sodium 王citrate	-	40 nm	-	[65]
	Au	Citrate, deionized water	hydroquinone	-	50–200 nm	-	[66]
-	Ni	deionized water	Na_2_CO_3_	4.29 m^2^/g	1.28–2.55 µm	-	[56]
-	V_2_O_5_	ethylene glycol, deionized water	-	-	2–3 µm	-	[50]
W/O microemulsion approach	CdSe	n-octane, 1-butanol, deionized water	CTAB	13.14 m^2^/g	2.5–3.5 µm	-	[57]
thermal oxidation	ZnO	-	-	-		photocatalytic	[58]
microwave-assisted method	TiO_2_	toluene	-	-	2–3 µm	photocatalytic	[60]
self-assembly	Polyaniline	deionized water	-	24.5 m^2^/g	2.5 µm	electrorheological	[61]
		ethanol	-	-	10 µm	electrochemical	[63]

**Table 2 materials-15-02846-t002:** The properties of urchin-like composite micro-/nanoparticles.

Material [Ref.]	Inner Diameter	Outer Diameter	Template	BET Surface Area	Application	Reference
ZnO	3 µm	4.3 µm	Polystyrene microsphere	-	-	[82]
	-	5–6 µm	glucose monohydrate	36.1 m^2^/g	EM wave absorption	[83]
	40 µm	50 µm	-	-	-	[80]
	1 µm	1.8 µm	-	-	-	[86]
	2 µm	4 µm	H2	-	solar cells	[93]
Polyaniline	280 nm	400 nm	Hollow polystyrene microsphere	-	-	[85]
	-	1.5 mm	sulfonated polystyrene microsphere	-	Electrochemical Energy Storage	[92]
TiO_2_	600 nm	1 µm	-	230 m^2^/g	Photocatalysis	[87]
	-	3 µm	O_2_	251.2 m^2^/g	electrorheological	[95]
TiO_2_@Ag	200 nm	600 nm	SiO_2_	-	surface-enhanced Raman scattering sensor	[84]
Fe_2_O_3_	600 nm	0.9 µm	CO, CO_2_	30.68 m^2^/g	-	[94]
Fe_3_O_4_ @PDA-Ag	200 nm.	350 nm	-	48.04 m^2^/g	catalytic	[91]
α-MnO_2_	1.4 µm	2 µm	-	132 m^2^/g	-	[88]
Gold	23–45 nm	104 nm	Ag nanoparticle		-	[89]
γ-Al_2_O_3_	-	2.5 µm	P123	210.2 m^2^/g	-	[96]
Co_3_O_4_	1–2 µm	5–8 µm	-	-	Lithium-ion batteries-	[90]
	800 nm	1.0 µm	-	-	Lithium-ion batteries	[99]
MoS_2_/NiCo_2_S_4_@C	500 nm	2 µm	Molybdenum-Glycerate nanospheres	100.31 m^2^/g	electrode	[98]
V_2_O_5_	670–730 nm	3–4 m	-	-	photodetector	[97]

**Table 3 materials-15-02846-t003:** The properties and application of urchin-like composite micro-/nanoparticles.

Material	BET Surface Area	Particle Size (Diameter)	Solvent	Specific Capacitance	Application	Ref.
BU-TiO_2–X_/Ag_3_PO_4_	-	-	deionized water	-	Photocatalytic, antibacterial	[67]
MgCo_2_O_4_@polypyrrole	-	9–12 µm	deionized water	1079.6 F/g at 1 A/g	supercapacitor	[68]
ZnO/Au/graphitic 王carbon nitride	45.2 m^2^/g	5 µm	deionized water	-	photocathodes	[69]
ZnO/TiO_2_	9.5 m^2^/g	5 µm	-	-	photocatalytic	[71]
carbon	159.5 m^2^/g	550–630 nm	deionized water	230 F/g at 0.5 A/g	supercapacitors	[70]
NiO-NiCo_2_O_4_	-	5 µm	deionized water		Li–O_2_ batteries	[73]
Bi_2_S_3_/Ag		2 µm	DMF		photocatalysts	[75]
metal/WO_3_	102 m^2^/g	1.5 µm	-		photocatalytic	[74]
Al-doped MnO_2_	-	-	deionized water	101 F/g at 5 A/g	-	[76]

Abbreviations: CTAB: cetyltrimethylammonium bromide; PEG-2000: polyethylene glycol 2000; DMF: N,N-dimethylformamide; SDS: sodium dodecyl sulfate; P123: (PEO)_20_–(PPO)_70_–(PEO)_20_; EG: ethylene glycol.

## Data Availability

Not applicable.

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
