# Peer review of "Progress in Preparation of Sea Urchin-like Micro-/Nanoparticles"

_materials, 2022, doi:10.3390/ma15082846_

Round 1
Reviewer 1 Report
The article submitted by R. Ma et al., titled "Progress in Preparation of Sea Urchin-like Micro/Nano Particles" describes a summary of the synthesis (by hydrothermal, thermo-solvent, electrochemical and microwave-assisted methods), application (photocatalysis, electrochemistry, electromagnetic wave absorption, electrorheological fluids and gas sensor) and reaction mechanism of urchin-like micro/nano particles with controllable structure and morphology.
In general, I strongly advise the authors to consider strengthening the review's structure and coherence, as well as the language and sentence structures, as well as providing a more detailed discussion of the introduction, examples, and outlook. Items in each paragraph appear to be merely handy mentions or regurgitation of well-known information, with no apparent value added in terms of establishing a context for the remainder of the content presented in the article. In the actual state, the review is not suitable for publication in MDPI-Materials journal. Nonetheless, I encourage the authors to re-submit this work.
Even though few reviews have been written in this field, the one presented here adds little value to what is known in the field and fails to address gaps that other similar reviews have not addressed. Along with the general points raised previously, the following are some additional points that require attention (see PDF attached):
- The language is in need of improvement. Numerous sentences contain syntax errors or are awkwardly structured. Numerous sentences have unclear meanings as well.
- The "conclusion and future outlook" section fell short of providing a comprehensive overview of the future opportunities and current challenges associated with these nanoparticles.
- Certainly, the text should be revised. There is an absence of thought flow between paragraphs.
- Some references in the field are missing, for example: https://pubs.acs.org/doi/abs/10.1021/cm060681i https://pubs.acs.org/doi/abs/10.1021/jp809224j
https://pubs.acs.org/doi/abs/10.1021/jp1119074

Reviewer 2 Report
Dear Authors,
Thank you for the interesting work.
I have some comments on it:
- Abstract has to be improved to give readers more a detailed information on the review's main outlines. At the moment, I think, Lines 23-24 ("the research progress of the preparation...") do not add anything to the phrase given in Lines 19-20 ("we summarize the synthesis...").
- I recommend to complete the Sections "2. Preparation..." and "3. Applications" with systematization using tables and/or original schemes made by the Authors. Otherwise it is not obvious what is the reasons of the review.
- Additionally, before section "Conclusions" it is need to write the discussion on the current state in the urchin-like structures technology and applications, reflecting the Authors' own point of view.
- Figure captions should have generalizing phrase, e.g. "Figure 1. Electron microscopy results: (a)... (b)...", but not to be a collection of unrelated illustrations ("Figure 3. (a)Preparation process... (b)Schematic illustration...").
- The figures' resolution has to be improved to make the text readable.
- Combined figures contain too small panels, e.g. Figure 3, Figure 4, Figure 5. I recommend to split the figures or to exclude some of them.
- Figure captions have to contain the copyright holders' information in addition to the references, e.g. "Copyright Elsevier, Wiley or Springer-Nature".
- What do the Authors mean by term "anti-spinel crystal", Line 165? If it is an inverse spinel, I recommend to use this standard term.
- The manuscript contains a lot of typos, e.g. Line 52, "structure[18,19],dendritic structure[20,21] ,urchin-like..."; Line 230, "γ -MNS crystal particles."; Lines 268-269, "Liu et al.[47] 。"; Line 426, "particle .The"; Line 427, "4a: Firstly"; Line 510, "Fig. 5c.."; "Line 793, "about 0.8 μm。"; Line 847, "(â…¢)Co9S8"; Line 937, "14.3 GHZ".
Round 2
Reviewer 1 Report
There has been an improvement, especially in the applications. However, the manuscript still needs work. There is still an absence of thought flow between paragraphs, mainly in the first half of the paper.
Please see the attached PDF file.

Reviewer 2 Report
Dear Authors,
Thank you for addressing all my comments. I believe now the manuscript is suitable for publication.
Author Response
Thanks for reviewer's positive recommending.